# CAM6 simulation of mean and extreme precipitation over Asia: Sensitivity to upgraded physical parameterizations and higher horizontal resolution

Lei Lin[1], Andrew Gettelman[2], Yangyang Xu[3], Chenglai Wu[4], Zhili Wang[5], Nan Rosenbloom[2], Susan C. Bates[2], Wenjie Dong[1]

[1]School of Atmospheric Sciences and Guangdong Province Key Laboratory for Climate Change and Natural Disaster Studies, Sun Yat-sen University, Zhuhai, Guangdong, China
[2]National Center for Atmospheric Research, Boulder, Colorado, USA
[3]Department of Atmospheric Sciences, College of Geosciences, Texas A&M University, College Station, Texas, USA
[4]International Center for Climate and Environment Sciences, Institute of Atmospheric Physics, Chinese Academy of Sciences, Beijing, China
[5]State Key Laboratory of Severe Weather and Key Laboratory of Atmospheric Chemistry of CMA, Chinese Academy of Meteorological Sciences, Beijing, China

*Correspondence to*: Wenjie Dong (dongwj3@mail.sysu.edu.cn)

**Abstract.** The Community Atmosphere Model version 6 (CAM6) released in 2018 as part of the Community Earth System Model version 2 (CESM2), is a major upgrade over the previous CAM5 that has been used in numerous global and regional climate studies. Since CESM2/CAM6 will participate in the upcoming Coupled Model Intercomparison Project phase 6 (CMIP6) and is likely to be adopted in many future studies, its simulation fidelity needs to be thoroughly examined. Here we evaluate the performance of a developmental version of the Community Atmosphere Model with parameterizations that will be used in the version 6 (CAM6α) with a default 1º horizontal resolution (0.9º × 1.25º, CAM6α-1º) and a high resolution configuration (approximately 0.25º, CAM6α-0.25º), against various observational and reanalysis datasets of precipitation over Asia. CAM6α performance is compared with CAM5 at default 1º horizontal resolution (CAM5-1º) and a high-resolution configuration at 0.25º (CAM5-0.25º). With the prognostic treatment of precipitation processes and the new microphysics module, CAM6α is able to better simulate climatological mean and extreme precipitation over Asia, to better capture the heaviest precipitation events, to better reproduce the diurnal cycle of precipitation rates over most of Asia, and to better simulate the probability density distributions of daily precipitation over Tibet, Korea, Japan and Northern China. Higher horizontal resolution in CAM6α improves the simulation of mean and extreme precipitation over Northern China, but the performance degrades over the Maritime continent. Moisture budget diagnosis suggests that the physical processes leading to model improvement are different over different regions. Both upgraded physical parameterizations and higher horizontal resolution affect the simulated precipitation response to internal variability of the climate system (e.g. Asia monsoon variability, ENSO, PDO), but the effects vary across different regions. For example, higher horizontal resolution degrades the model performance in simulating precipitation variability over Southern China associated with the East Asia summer monsoon. In contrast, precipitation variability associated with ENSO improves with upgraded physical

parameterizations and higher horizontal resolution. CAM6α-0.25º and CAM6α-1º shows an opposite response to the PDO over Southern China. Basically, the response to increases in horizontal resolution is dependent on CAM version.

## 1 Introduction

The Community Atmosphere Model (CAM) is an atmospheric general circulation model (AGCM) developed at the National Center for Atmospheric Research (NCAR) with extensive community support. The fifth version of CAM (CAM5) [*Neale et al*., 2010], as part of the Community Earth System Model version 1 (CESM1) [*Hurrell et al*., 2013], was widely used for climate studies over Asia. CAM5 included a two-moment cloud microphysics scheme that is missing in the previously CAM versions and improved the representation of low-level clouds [*Morrison and Gettelman*, 2008], net conversion rates from water vapor to cloud condensation [*Neale et al*., 2010; *Zhang et al*., 1995], and a three-mode aerosol module [*Liu et al*., 2012].

Among various applications, CESM1/CAM5 has been used in many studies relating to clouds and precipitation over Asia. For example, *Zhang et al*. [2014] evaluated the sensitivity of simulated stratus clouds over eastern China to horizontal resolution in CAM5. *Zhang and Chen* [2016] investigated the mean state and diurnal cycle of summer precipitation over continental East Asia in CAM4 and CAM5. *Li et al*. [2015] used CAM5 with different resolutions (2.8º, 1.0º and 0.45º, respectively) to study the impact of horizontal resolution on model performance in simulating precipitation over East Asia. *Wang et al*. [2018] investigated the sensitivity of the Indian Summer Monsoon to different convective schemes in CAM5. *Jiang et al*. [2015] examined anthropogenic aerosol optical depths and their effects on clouds and precipitation in East Asia. *Vinoj et al*. [2014] found with CAM5 simulation that dust-induced atmospheric heating over North Africa and West Asia modulated monsoon rainfall over central India.

A pre-released version of CAM6 (denoted here as CAM6α, extensively tested in late 2017) shares the same basic physics as the released version of CAM6, except for slightly different tuning parameters and without the updated surface drag scheme. CAM6α was thoroughly evaluated over the continental United States [*Gettelman et al*., 2018], but the improvement of CAM6α over its predecessors remains unknown for East and South-East Asia. Since CAM6 and other CMIP6 (Coupled Model Intercomparison Project phase 6) [*Eyring et al*., 2016] era models are likely to be used widely for hydroclimate studies over Asia in the next five to ten years, model fidelity needs to be carefully evaluated. For example, the UK Met Office atmosphere model Global Atmosphere 6.0 (GA6) was used to study the interannual and intraseasonal precipitation variability over China [*Stephan et al*., 2018a and b; *Walters et al*., 2017]. Similarly, *Martin et al*. [2017] analyzed tropical precipitation in GA6 with a range of horizontal resolutions and found that the behaviour of the deep convection parameterization in GA6 is largely independent of the grid-box size and time step length over which it operates.

In addition to model physics upgrades, another area of growth in global climate model development is the enhancement of horizontal spatial resolution. 0.25º (approximately 25 km) grid spacing is the targeted resolution for global atmosphere modeling in the near future [*Sharma et al*., 2016], while most of default CMIP6 runs at decadal to centennial

scale are still performed at 1º. Generally, enhanced model resolution tends to reduce model biases [*Palmer*, 2014; *Yao et al.*, 2017; *Chen et al.*, 2018]. Nevertheless, *Johnson et al.* (2016) indicated that increasing horizontal resolution was not a solution to many South Asian monsoon biases in the Met Office Global Atmosphere 3.0 (GA3) model. For CAM, high-resolution performance has been evaluated in several studies. *Wehner et al.* [2014] found that the extreme precipitation amounts are larger as the resolution increase in CAM5. Other CAM5 studies have also tested resolutions of 0.5º [*Bacmeister et al.*, 2013; *Lau and Ploshay*, 2009] or 1º [*Zhang et al.*, 2014; *Li et al.*, 2015; *Lin et al., 2015*; *Lin et al.*, 2016]. However, the performance of 0.25º CAM6 simulations over Asia has not been examined rigorously.

In order to provide insights into both physical schemes and horizontal resolution, here we analyze a 4-model hierarchy. First, we explore the effects of new physical parameterizations by contrasting a CAM6α-1º simulation with CAM5-1º. Second, we evaluate CAM6α with 1º and 0.25º resolutions to quantify model sensitivity to horizontal resolution. In addition, we also analyse CAM5 at 0.25º and 1º resolution to evaluate if the impact of higher resolution is model dependent.

This technical paper of model evaluation is divided into the following sections. After the introduction (Section 1), Section 2 provides detailed information on the model configurations, model experimental set up, as well as observational datasets used as benchmarks. In Section 3, we compare the climatological (multi-year) average of monthly mean and daily extreme precipitation simulated by four versions of CAM in the context of observational uncertainty. Section 4 is devoted to the evaluation of precipitation variability at a wide range of time scales that is not tuned during the model development process. Section 5 discusses in detail whether climate simulations are improved over Northern China, South-western China and the Maritime continent, due to new physical parameterizations and/or higher resolution (by conducting a moisture budget analysis using both CAM5 and CAM6). Finally, we present further discussion and a summary in Section 6.

## 2 Methods

### 2.1 CAM simulation

The first set of simulations use the publicly released version of CAM5.1 [*Neale et al.*, 2010]. CAM5.1 treats stratus cloud microphysics with a double moment formulation of *Morrison and Gettelman* [2008] and *Gettelman et al.* [2008]. The spatial distribution of shallow convection is simulated with a set of realistic plume dilution equations [*Park and Bretherton*, 2009]. The ice cloud fraction scheme allows supersaturation via a modified relative humidity over ice and inclusion of ice condensation amount [*Gettelman et al.*, 2010]. A three (log normal) mode aerosol model was used to predict aerosol concentration, and the number concentration of aerosols are connected to ice/warm cloud microphysics accounting for ice and liquid activation of cloud crystals and drops [*Liu et al.*, 2012]. A Finite-Volume (FV) dynamical core (1º) is used. The second set of simulations use CAM5 with the Spectral Element (SE) dynamical core at a resolution of 0.25º [*Meehl et al.*, 2019, to be submitted]. The other settings are identical to the first set of simulations of CAM5-1º.

The third and fourth sets of simulation considered in this study use a near-final version of CAM6 (denoted as CAM6α) that has the same basic physics as the final version of CAM6 released in 2018 [*Bogenschutz et al.*, 2013; *Gettelman et al.*,

2018] but with slightly different tuning parameters. CAM6α uses CLUBB (Cloud Layers Unified By Binormals) [*Golaz et al.*, 2002a, b] to unify the boundary layer and shallow convective turbulence with cloud macrophysics, a new ice nucleation parameterization [*Hoose et al.*, 2010], updated cloud microphysics with prognostic precipitation [*Gettelman et al.*, 2015; *Gettelman et al.*, 2018], and modified aerosol modal model [*Liu et al.*, 2016]. CAM6α, with the Spectral Element (SE)

dynamical core [*Lauritzen et al.*, 2018], was run twice with different horizontal resolutions (i.e., ~1º and ~0.25º). The same uniform resolution SE simulations with CAM6α forced by observed sea surface temperature data from 1979-2005 (saved output for monthly, daily and 3 hourly) are analyzed in *Gettelman et al.* [2018]. Using these simulations, we can evaluate the differences due to resolution increase from 1º to 0.25º in CAM5 (set 1 & 2) and CAM6 (set 3 & 4). That offers us an opportunity to explore whether the response from higher resolution is dependent on CAM version.

All model simulations evaluated here (Table 1) followed the protocol of the Atmospheric Model Intercomparison Project (AMIP). Simulations are forced by observed monthly sea surface temperature and sea ice from 1979 to 2005, which are linearly interpolated to obtain specified daily values, as well as the evolution of aerosol emissions and trace gases concentration (including $CO_2$). The 25-year simulations from 1980-2004 are analyzed here to match with available observational and reanalysis datasets. Monthly and daily frequency output data are used.

**2.2 Observational datasets**

**2.2.1 Asian Precipitation - Highly-Resolved Observational Data Integration Towards Evaluation (APHRODITE)**

     For the direct observation of temperature and precipitation, APHRODITE data is used [*Yatagai et al.*, 2012]. APHRODITE is a gridded daily precipitation product covering monsoonal Asia, Middle East and Russia, and is available at 0.25º × 0.25º resolution during the period of 1951–2007. The dataset was created by collating a dense network of rain gauge

daily measurements across Asia.

**2.2.2 The Japanese 55-year reanalysis (JRA55)**

     The Japan Meteorological Agency (JMA) reanalysis dataset (JRA55) uses an operational data assimilation system with the 4DVar scheme [*Kobayashi et al.*, 2015]. The dataset covers 55 years from 1958 (when regular radiosonde observation began on a global basis) to 2013. JRA55 has a horizontal resolution of 0.56°.

**2.2.3 Modern-Era Retrospective analysis for Research and Applications version 2 (MERRA2)**

     MERRA2 is a NASA atmospheric reanalysis for the satellite era (from 1980 to present) using the Goddard Earth Observing System Model, Version 5 (GEOS-5) with its Atmospheric Data Assimilation System. MERRA2 has a spatial resolution 0.625º × 0.5º [*Gelaro et al.*, 2017].

     The APHRODITE data might be limited by the potential lack of gauge observations in mountainous areas [*Zhao et al.*,

2015], and therefore we adopted MERRA2 as an additional data source specifically for daily precipitation evaluation. Note,

however, that rainfall in the reanalysis products such as JRA55 and MERRA2 is dependent on the reanalysis model physics, and it is well-known that there is large uncertainty among various observational datasets [*Sheffield et al.*, 2012; *Trenberth et al.*, 2014].

### 2.2.4 Tropical Rainfall Measuring Mission (TRMM)

The latest TRMM 3B43 Version 7 data (0.25º) between 1998 and 2016, downloaded from NASA Goddard Space Flight Center, combines multiple independent precipitation estimates from TRMM Microwave Imager and Global Precipitation Climatology Project (GPCP) rain gauge analysis. The 3-hourly data over Asia (a temporal resolution unavailable in other three data sources) from 1999-2004 are used in our study to assess the diurnal cycle of precipitation [*Huffman*, 2013].

### 2.2.5 Climate Prediction Center (CPC) temperature

CPC global datasets for daily surface air temperature from in situ measurements with 0.5º × 0.5º resolution is used as a second benchmark [*Chen et al.*, 2008]. CPC data is provided by the NOAA/OAR/ESRL (National Oceanic and Atmospheric Administration/Oceanic and Atmospheric Research/Earth System Research Laboratory) PSD (Physical Sciences Division), Boulder, Colorado, USA.

### 3 Climatological statistics of mean precipitation and daily extreme precipitation

Figure 1 shows the annual mean surface air temperature differences relative to APHRODITE. First note that although sharing some common surface measurements, JRA55 and APHRODITE have major differences over the Tibetan Plateau and Southeast Asia regions (Figure 1a), while the CPC data shows large differences (relative to APHRODITE) over Sichuan province in southwestern China (Figure 1b). We should not expect an atmospheric numerical model, which is only constrained at the surface by ocean temperature, to have a better agreement with the observational benchmark (APHRODITE here) than the reanalysis product (JRA55), which is fully constrained by both ground and atmospheric observations. Thus, only the regions over which the model-APHRODITE difference is larger than JRA55-APHRODITE difference are considered 'significant' and stippled in Figure 1c-f.

Similar to the difficulty in capturing the high-elevation temperature in JRA55, there is a warm bias over Eastern and Southern China and a cool bias over Tibet in the CAM5 simulations (Figure 1c). The cool bias over the Tibetan Plateau and warm bias over the foothill regions (both the Indian side and the northern edge of the Tibetan Plateau) are a long-standing bias in many global and regional climate models. We note that the bias over Sichuan province appears to be muted in the high-resolution version (Figure 1e and 1f), illustrating the promise of further enhancing the resolution.

Moreover, CAM6α reduces the temperature bias over Southern China (Figure 1c-d). Over the entire domain considered in Figure 1, the Root Mean Square Difference (RMSD) relative to APHRODITE is 2.3℃ for CAM5-1˚, 2.4℃ for

CAM6α-1˚, 2.2°C for CAM5-0.25˚ and 2.3°C for CAM6α-0.25˚ (Note that RMSD for JRA55 is 1.5°C). The RMSD as a regional average quantity is not sufficient in characterizing the performance of models that vary at finer scales. Because of this limitation, in the following evaluation for precipitation, we divide East Asia into several regional boxes (Figure 2).

Figure 2 illustrates the climatological precipitation biases relative to APHRODITE. MERRA2 is considered here as the third 'data' source in addition to JRA55 and APHRODITE, as an estimate of the large uncertainty of observational datasets [*Herold et al.*, 2015, 2016]. All four CAM simulations have a dry bias over Southern China and wet bias over the rest of China, especially the Sichuan basin (near the eastern edge of the Tibetan Plateau) and the Himalayan mountain range that defines the southern edge of the Tibetan Plateau. The precipitation biases in those regions are particularly important for two reasons: (a) many major rivers in East Asia, South Asia and Southeast Asia have their headwaters in those regions; (b) these regions are also prone to natural hazards such as landslides.

Overall, CAM6α-1º performance is slightly better than CAM5-1º, and CAM6α-0.25º falls in between the 1º resolution models (RMSD is 1.83 mm/day for CAM5-1º, 1.62 mm/day for CAM6α-1º and 1.71 mm/day for CAM6α-0.25º). The CAM5-0.25˚ RMSD (1.42 mm/day) is better than that of CAM6-0.25˚ due to the lower bias over Himalayan and Kalimantan (Figure 2e and 2f).

Because of large spatial heterogeneity, eight regions are selected to evaluate precipitation. Five domains are shown as colored boxes in Figure 2c: (1) Tibet: 27°N–37°N, 79°E–99°E; (2) Southwestern China: 28.5°N–35.5°N, 100°E–105°E; (3) Korea: 34°N–40°N, 124.5°E–129.5°E; (4) Japan: 31°N–43°N, 130°E–144°E; (5) the Maritime Continent: 9.75°S–19.75°N, 90°E–150°E. Only the land within these boxes is considered in this study. The other three are India, Northern China and Southern China. The "India" average is entirely within mainland India. "Northern China" and "Southern China", are also defined in Figure 2c, as the where "Northern China" north of the "Qin Mountain and Huai River" at 32.8°N, and "Southern China" south of this. The western boundary of "Northern China" and "Southern China" is a straight line named as "Hu-Huanyong Line" between Heihe (50.2°N, 127.5°E) and Tengchong (24.5°N, 98.0°E).

Figure 3 shows the RMSD and mean bias (taking the domain average of a respective model simulation or data products minus APHRODITE) of annual mean precipitation for nine selected regions. Note that over a few selected regions, the model performance is as good as reanalysis (such as over Japan, Korea and Southern China) and thus we will not further investigate model improvements. Among regions where CAM models perform poorly compared with reanalysis (Southwestern China, Tibet, India, Northern China and Maritime continent), CAM6α-1˚ performance (RMSD) is better (lower) than CAM5-1˚ for Tibet, Southwestern China and Northern China, but gets worse for the Maritime continent. Notably, CAM6α-0.25˚ is closer to observations over Southwestern China and Northern China, but also gets considerably worse for the Maritime continent (with a large RMSD=7.2 mm/day and bias=4.0 mm/day). Both CAM versions with higher resolution simulate the climatological precipitation over Northern China better. Increasing model resolution decrease the RMSD and bias of CAM5 over Tibet and Southwestern China while increase those of CAM6 (Figure 3).

We will explore the details that might lead to the progressive improvement over Southwestern China and Northern China and the poor performance of CAM6-0.25˚ over Maritime continent in Section 5.

We next assess the model performance in simulating convective and large-scale precipitation components separately. The ratio of convective to large-scale precipitation is a useful diagnostic, because both convective activity and large-scale instability can lead to precipitation in this model. Most atmospheric models use convective parameterizations to represent the effects of sub-grid scale convective processes, with reduced complexity microphysics [*Zhang and McFarlane*, 1995; *Kooperman et al.*, 2016]. The convective precipitation in CAM5 includes the shallow and deep convective precipitation. In CAM6 the shallow convective regime is handled by CLUBB coupled to stratiform microphysics and hence shallow convective precipitation is prognostic and part of the large-scale precipitation.

The latent heating required by the atmosphere imposes an important constraint on the amount of mean rainfall, but model horizontal resolution and parameter settings dictate the time scales [*Gustafson et al.*, 2014]. CAM6 estimates shallow convective precipitation from the prognostic calculations in CLUBB (which have memory between timesteps of turbulent motion) rather than diagnostically representing the effects of sub-grid scale convective processes at each location and timestep. One of the reasons why the simulated rainfall intensity is expected to improve when the model is run at higher resolution is that the variance of sub-grid scale humidity and thermodynamics drops, and the parameterized sub-grid scale processes (such as sub-grid scale turbulence with CLUBB) are better separated into regimes [*Kopparla et al.*, 2013].

Figure 4 illustrates the ratio of convective (PRECC) to large-scale (PRECL) precipitation. This ratio (PRECC/PRECL) is greater over the ocean than over the land, as expected. CAM6α-1˚ has a larger ratio over the tropics, compared to CAM5-1º. CAM6α-0.25˚ simulated a lower ratio than CAM6α-1˚ (Figure 4b) and a similar pattern with CAM5-0.25˚ (Figure 4c).

Higher horizontal resolution models tend to simulate higher vertical velocities [*Gettelman et al.*, 2018] and a lower ratio of convective to total rainfall. A larger fraction of precipitation can be resolved as the consequence of large-scale flow, limiting the need to invoke sub-grid convective schemes. Besides, increasing resolution better resolves topographic and surface effects, and separates regimes as sub-grid scale variance is reduced, particularly in the thermodynamic variables. With a fixed amount of precipitable water, more condensation caused by the stratiform scheme means less is available for convective precipitation, so the ratio becomes lower.

The compensation above is a feature of the physical parameterization suite in CAM due to timescale. Large-scale liquid condensation by the resolved scale cloud schemes (CLUBB and microphysics), instantaneously condenses all vapor in excess of liquid saturation to cloud liquid. In contrast, the deep convective parameterization has a timescale that produces mass flux and precipitation at a defined rate. Note that as the timestep gets shorter, the mass flux and precipitation over a timestep will decrease, which is the major reason for the decrease in deep convective precipitation in CAM6. The large-scale condensation (including shallow convection and cloud microphysics) does more as the time step changes, while the deep convective parameterization does less [*Gettelman et al.*, 2018].

Figure 5 uses Taylor diagrams [*Taylor,* 2001] to evaluate the simulation of annual and seasonal surface air temperature, mean precipitation and two metrics of extreme precipitation. Red circles show the results from CAM5-1º, blue for CAM6α-1º and green for CAM6α-0.25º. Movement closer to the point of (1.0, 1.0) in the Taylor diagrams indicates improvement of the simulation relative to APHRODITE. For example, the model circles are very close to the blue circles representing JRA55

for near-surface (2m) temperature (TREFHT) (Figure 5a), which indicates good performance for surface air temperature for all three models. The annual and seasonal precipitation (PRECT) correlations of the two CAM6α simulations are around 0.8, while the annual correlation of CAM5-1º is 0.7 (Figure 5b). CAM6α-0.25º Root Mean Square (RMS) is bigger than that of CAM6α-1º (green circles relative to blue circles in Figure 5b). The maximum daily precipitation during each month of a year (RX1day) of CAM6α-1º performs better than CAM5 (blue circles relative to red circles in Figure 5c). The number of heavy precipitation days (R10) has large differences among the observational and reanalysis data (black and gold circles in Figure 5d). CAM6α have higher correlation coefficients with APHRODITE (~0.8) than CAM5 (~0.7).

Overall, CAM6α models perform better than CAM5-1º for the mean precipitation (Figure 5b), maximum daily precipitation and number of heavy precipitation days values (Figure 5c and d). Although CAM6α-0.25º performs worse than 1º for the mean precipitation and RX1day (green circles relative to blue circles in Figure 5b and 5c) mostly due to the bias over the Maritime Continent (defined as: 9.75°S–19.75°N, 90°E–150°E), CAM6α-0.25˚ performs better for R10 (Figure 5d). All three model versions show that precipitation bias in JJA is bigger than other seasons (Figure 5b and 5c).

## 4 Variability of simulated precipitation over Asian regions in various time scales

In this section, we evaluate simulated rainfall variability across a wide range of time scales. This serves as a useful test of model performance, because it is difficult to improve model performance in simulating temporal variability, as opposed to the climatological average, based on simple tuning of one or more parameters. Moreover, it is expected CAM6 will be widely used to study climate variability at various time scales.

### 4.1 Variability of precipitation due to PDO, ENSO and East Asia summer monsoon index

The Pacific Decadal Oscillation (PDO) represents variability in the tropical and extratropical North Pacific at interdecadal time scales that significantly impact climate [*Mantua et al.*, 1997; *Meehl et al.*, 2013]. Figure 6 shows a regression of observed and simulated annual precipitation against the PDO index, derived from the leading principal component of monthly SST anomalies in the North Pacific Ocean [*Mantua et al.*, 1997]. A common feature as revealed in the observational records is the drying tendency over Indochina (e.g., Thailand) during positive PDO phases. This feature is well represented in all four CAM versions. Similarly, a wet anomaly over most of India during positive PDO is also captured in all CAM versions. In contrast, the wet anomaly over Southern China is simulated well in the CAM6-0.25º version, while the opposite anomaly is seen in the CAM6α-1º and CAM5-0.25º.

The El Niño / Southern Oscillation (ENSO) plays a central role in ocean-atmosphere coupled interannual variability. CAM5 has been widely used to study ENSO impacts over Asia [*Chen et al.*, 2018; *Hoell et al.*, 2016]. Figure 7a-g shows the regression coefficients between annual mean precipitation and ENSO. The ENSO index is the cold tongue index following *Deser and Wallace* [1987]. Both 1º CAM5 and CAM6α (Figure 7d and e) capture the observed wet anomaly over Pakistan and Afghanistan and dry anomaly over Indonesia. However, we find that the drying tendency over Southern China during El Nino years (the upper row of Figure 7) is completely missing in CAM5 but starts to emerge in CAM6α-1º version and gets

better in CAM6α-0.25º version. The influence of ENSO on boreal summer monsoon rainfall over Asia is well known. Figure 7h-n shows the regression coefficients between JJA precipitation and ENSO. Those patterns are similar when the analysis is done with annual mean precipitation but with a weaker correlation of annual precipitation and ENSO. Our results here thus call into questions of the fidelity of previous ENSO studies on hydroclimate over Southern China using CAM5.

5          Next, we examine seasonal precipitation variability associated with the East Asia Summer Monsoon (EASM). The EASM is an important climate system over Eastern China. Climate models are widely used for EASM studies [*Zhou and Li*, 2002; *Chen et al.*, 2010; *Zhou et al.*, 2013], although model biases for the EASM are a long-lasting problem from CMIP3 (Coupled Model Intercomparison Project phase 3) to CMIP5 with limited improvement [*Song and Zhou*, 2014; *Kusunoki and Arakawa*, 2015]. The East Asia Summer Monsoon Index (EASMI) is a unified Dynamic Normalized Seasonality (DNS) 10 monsoon index defined by *Li and Zeng* [2002, 2003]:

$$EASMI = \frac{\left\| \overline{V_1} - V_i \right\|}{\left\| \overline{V} \right\|} - 2 \, , \qquad\qquad (1)$$

Where $\overline{V_1}$, $V_i$ are the January climatological and monthly wind vectors for a grid, respectively, and $\overline{V}$ is the mean of January and July climatological wind vectors for the same gird. The constant 2 on the right-hand side of the formula is the determinant criterion. The double vertical line indicates the normalized value. EASMI can be used to depict both the 15 seasonal cycle and inter-annual variability of the EASM. The EASMI summer average is considered here.

          Figure 8 shows regression coefficients between summer (JJA) precipitation in the East of China and EASMI. There is an apparent negative relationship between the EASMI and summer rainfall in the middle and lower reaches of the Yangtze River in China for the observational and reanalysis data (Figure 8a-c). All three CAM models capture the positive correlation over Southern China (Figure 8d-g) found in the observations and reanalysis (Figure 8a-c), although CAM6α at both 1˚ and 20 0.25˚ and CAM5-0.25˚ has the northern edge of the positive correlation more northward (about 2º latitude) distributed than that of observational and reanalysis data (Figure 8e-g).

## 4.2 Seasonal cycle

          Due to the monsoonal influence on both East Asia and South Asia, summer (JJA) precipitation dominates the annual total. Thus, the summer-winter contrast in precipitation serves as a useful quantity for model evaluation. JRA55 has a less 25 summer-winter contrast than APHRODITE (Figure 9a). Compared to APHRODITE, MERRA2 has almost the same annual cycle over China (Figure 9b). In CAM5, the JJA precipitation appears to be too large compared to the rest of the year. CAM6α-1º simulations reduce the bias in the annual cycle over Pakistan (Figure 9c and 9d), although models continue to overestimate seasonal variability over arid regions in western China. CAM6α-0.25˚ shows a bigger bias than CAM5-0.25˚ over the Himalayas (Figure 9e and 9f). The seasonal variability is significantly improved in CAM6α-0.25º, especially in

Himalayan mountain regions. Southern China, again, is a region where the CAM6-0.25º's performance is improved substantially. This is further diagnosed in the next Figure.

In Figure 10, we show the time series of zonal mean precipitation between 100ºE and 125ºE to evaluate model performance in simulating the annual cycle of EASM-related precipitation. Figure 10 shows that precipitation mainly occurs from May to September, and rapidly shifts to Northern China around June and continues to September [*Su et al*., 2017]. It is clear that APHRODITE, JRA55 and MERRA2 all depict such a northern shift (Figure 10a, b and c). Both versions of CAM6α capture this northern shift and its persistent from June to September, especially CAM6α-0.25º (Figure 10e and f), while CAM5-1º illustrates persistence from June to July only (the area of continuous yellow shading in Figure 10d less than those of CAM6). High resolution CAM5-0.25º captures that shift and its persistence from June and September (Figure 10g). This explains the CAM5-1º deficiency as shown in Figure 9.

## 4.3 Daily precipitation frequency and diurnal cycle of precipitation

Figure 11 illustrates the daily rainfall frequency distribution from reanalysis/observation and models. Despite large uncertainties among observational datasets, daily rainfall intensity for CAM6-0.25˚ (green line) is closer to observational values over Southwestern China than other simulations (Figure 11b). CAM5-0.25˚ overestimates the frequency of light precipitation (0.1-10 mm/day) over Southwestern China (purple line in Figure 11b). CAM5-1˚ simulates a higher frequency of light precipitation (0.1-1.0 mm/day) over Korea, Japan, Northern China and Southern China (Figure 11a, c, d, g and h). New physical schemes in CAM6-1º capture the observed distribution over Korea and Japan (Figure 11c and d). Realistic rainfall intensity is not energetically necessary because more frequent weak events can produce the same latent heating as less frequent but more intense rainfall events. CAM6 improves the light large-scale precipitation over Korea, Japan and Northern China (not shown).

To look into the heavy precipitation more carefully, Figure 12 shows precipitation percentiles from 90% to 99.99%, which captures the heaviest precipitation events [*Kooperman et al*., 2016]. CAM6 with new physics modules (blue) has better performance than CAM5 over five of eight selected regions (Tibet, Southwestern China, Japan, India, Northern and Southern China). Higher horizontal resolution in CAM6 and CAM5 (green and purple) simulates better intensities over the Maritime Continent (Figure 12e), but the results of CAM6 (CAM5) degrade (upgrade slightly) for the heaviest precipitation events over India, Northern and Southern China (Figure 12f-h). No significant differences are found between CAM5 and CAM6 over Tibet, Southwestern China, Japan and the Maritime Continent.

Figure 13 shows diurnal cycles of June precipitation rate from TRMM satellite observations and model simulations. Satellite observations show that the diurnal peak in precipitation is around 20:00 (local time) over Tibet, India and the Maritime continent, while the peak over Southern China is in the afternoon (15:00 local time) (Figure 13a). CAM6α reproduces many features of the TRMM observations (Figure 13c and 13e), while CAM5-1º and CAM5-0.25˚ simulates an earlier-in-the-day peak precipitation over India, Southern China and Maritime continent (Figure 13b and 13d). CAM6α-0.25º simulates the diurnal cycle better than CAM6α-1º in southern Tibet (Figure 13d) along the Himalayas. In general, CAM6 at

all resolutions has significant improvements over CAM5 in simulating the diurnal cycle of precipitation in East Asia, CAM5 with higher resolution improves the diurnal cycle only slightly.

The CLUBB parameterization appears to be the reason for this improvement. CLUBB has prognostic moments that provide memory that facilitate the initiation of shallow convection in the mid-morning and early afternoon. It is important to note that improved simulation of the diurnal cycle by CLUBB is a robust feature in every coupled and atmosphere-only simulation with CAM6α. The CLUBB unified parameterization is able to prevent the deep convective scheme from firing off too early and better simulate a gradual transition of these regimes successfully [*Bogenschutz et al.*, 2018].

**5 Change due to physical parameterizations and high resolution: Northern China, Southwestern China and the Maritime Continent**

Here, we quantitatively contrast climate variables over Southwestern China, Northern China and the Maritime Continent, since the model with new physical modules or high horizontal resolution simulates a better climatology over Southwestern China and Northern China but does the opposite over the Maritime Continent (Figure 3). We attempt to attribute changes to either physical parameterizations or resolution. Additionally, we investigate whether the improvement due to resolution is dependent on CAM version. Table 2 illustrates simulation differences due to physical parameterizations (CAM6α-1˚ minus CAM5-1˚) and higher horizontal resolution (CAM6α-0.25˚ minus CAM6α-1˚ and CAM5-0.25˚ minus CAM5-1˚, respectively).

New physical modules and higher horizontal resolution in CAM6α perform better over Southwestern China by decreasing convective precipitation (by -0.38 and -0.81 mm/day, respectively). New physical modules simulate larger surface latent heat flux but less vertically-integrated humidity, which lead to a decrease in convective precipitation over Southwestern China (Table 2). Higher resolutions in CAM6 and CAM5 both decrease the surface latent heat flux and convective precipitation over Southwestern China (Table 2). However, higher resolution leads to an opposite change in surface sensible heat flux (total cloud amount) between CAM5 and CAM6α by 12.7 and -7.4 W/m$^2$ (-3.9 and 0.5 %), respectively.

Newer physics parameterizations in CAM6α simulate a stronger solar flux reaching the surface in Northern China (Table 2). This may be due to improvements in the diurnal cycle of precipitation [*Li et al.*, 2008]. The stronger solar flux leads to larger latent heat release, although convective precipitation over Northern China is not changed by new physics modules. Better representation of topography at 0.25˚ resolution leads to increased downward shortwave flux at the surface in Northern China. Less convective precipitation associated with increased resolution is seen over Northern China in Table 2 (-0.31 mm/day for CAM6 and -0.33 mm/day for CAM5). Note that CAM6 simulates a decrease in the surface latent heat flux with a higher resolution, while CAM5 does not. We also examine the moisture budget using diagnostics for precipitation changes [*Chou and Lan*, 2012]. The moisture budget analysis defines the mass conservation of water substance in an atmospheric column as:

$$\overline{P} + \overline{<\partial_x (uq)>} + \overline{<\partial_y (vq)>} + \mathrm{Re}\,s = \overline{E} \quad ,$$

(2)

where P is precipitation, q is specific humidity, u (v) is zonal (meridional) wind and E is evaporation into the atmosphere. <X> is a mass-weighted vertical integral and $\overline{X}$ denotes a temporal average. The horizontal advection can be further decomposed into the stationary and transient terms based on:

$$X = \overline{X} + X' = [\overline{X}] + \overline{X}^* + X' \quad ,$$

(3)

$[\overline{X}]$ ($\overline{X}^*$) are climatological zonal mean (stationary) eddy and $X'$ is transient variation. See *Yao et al.* [*2017*] and *Chen et al.* [*2018*] for more details.

No significant difference in evaporation (E) is seen between the model and JRA55 data (Figure 14). The results indicate that CAM6 (both resolutions) simulates a moisture budget closer to JRA55 than CAM5-1° and the model

precipitation bias appears mostly in the zonal moisture flux convergence term $(-\partial_x (\overline{uq}))$ over Southwestern China (Figure 14a). Although a large residual over Northern China (Figure 14b) may result from water vapor transport by the surface vertical movement induced by terrain slope [*Trenberth and Guillemot* 1995; *Seager et al.*, 2010], all four versions of CAM and JRA55 show that the zonal mean of specific humidity eddy transport $[\overline{q}]$ dominates Northern China precipitation (Figure 14b). CAM5-0.25° simulates similar moisture budget with CAM6-0.25°, while the corresponding results are different

between CAM5-1° and CAM6-1° over Southwestern China and Northern China (Figure 14a and 14b).

Next, we explore the differences in simulated variables due to higher horizontal resolution over the Maritime continent (Figure 3). As seen in Table 2, the higher horizontal resolution in CAM6 not only increases the vertically-integrated total cloud cover over the Maritime continent, but also leads to more shortwave flux reaching the surface, which tends to release more latent heat. Both CAM5 and 6 versions with 0.25° resolution reduce the convective precipitation and

20 increase the large-scale precipitation relative to 1° resolution, which leads to overestimation of total precipitation. The two 0.25° resolution CAM versions simulate the same surface air temperature, surface energy terms (surface latent and sensible heat flux, downwelling solar flux and longwave flux at surface) and vertically-integrated humidity change but simulate a different vertically-integrated total cloud change and downwelling clear-sky solar flux at surface (Table 2). With a fully coupled Community Climate System Model Version 4 (CCSM4), an earlier version of CAM (CAM4), *Shields et al.* [2016]

have shown that higher horizontal resolution tends to decrease convective precipitation and increase large-scale precipitation. The moisture budget analysis shows that the meridional specific humidity eddy transport is the main factor leading to the bias over Maritime continent (Figure 14c). Note that the analysis in this study is focused on the Maritime continent land area only. *Johnsan et al.* [2016] carried out a moisture budget analysis for the whole Maritime continent (including the ocean) and found that increased resolution causes increased moisture convergence and precipitation on the windward (southern)

side of the orography, which leads to decreased moisture availability on the leeward (northern) side, reducing precipitation.

## 6 Summary

Here we presented a comprehensive evaluation of AMIP-style experiments from 1980 to 2004 using the Community Atmosphere Model version 5 and 6 prototypes (CAM5 and CAM6α) at 0.25° and 1° spatial resolutions. CAM6 is very different from CAM5, because it has included a unified higher order closure scheme for the boundary layer, shallow convection and cloud macrophysics. Other updates in CAM6 include prognostic precipitation in the microphysics, a four-mode aerosol model, and ice nucleation schemes. In more conventional physical parameterization suites, shallow convection, cloud macrophysical parameterization schemes and the planetary boundary layer may not be compatible with one another. The CLUBB parameterization in CAM6, instead, represents a "unified" parameterization that is responsible for boundary layer processes, warm cloud macrophysics and shallow convective processes. The explicit representation of shallow convection precipitation is interactively coupled to stratiform microphysics within CLUBB, which replaces the previous convective parameterizations that diagnose the large-scale instability and the shallow convective response separately.

The major findings of this study are:

(1) For the climatology of mean precipitation over Asia, CAM6α-1° substantially reduces model biases. Using an observational dataset (APHRODITE) as the baseline, the mean squared error is reduced by 11% in CAM6α-1° compared to CAM5-1°. In terms of spatial distribution, the most remarkable bias of CAM5 and CAM6 is excessive precipitation in central China and a rainfall deficit over South China. Higher resolution in CAM5 and CAM6 both decrease the mean squared error.

(2) CAM6α better simulates the probability distributions of daily precipitation over Tibet, Korea, Japan and Northern China. Specifically, CAM6α performs better for the frequency of daily light precipitation over Korea, Japan and Northern China and captures the heaviest precipitation events over most of Asia. Notably, CAM6α models capture the observed diurnal cycle of precipitation. With a prognostic treatment of large-scale instability and the convective response, higher horizontal resolution in CAM6 leads to better performance on the frequency distributions of daily precipitation over Southwestern China (close to the edge of Tibet Plateau) and heavy precipitation over Northern China. The improvement, however, is dependent on CAM versions.

(3) Upgraded physical modules in CAM6 decrease the simulated convective precipitation and reduce the total precipitation bias over Southwestern China. More complex terrain at higher resolution leads to lower convective precipitation at higher resolution for both CAM versions, which improves the model performance in the total precipitation over Southwestern China and Northern China. Those improvements can be explained in terms of the meridional moisture convergence or zonal mean humidity eddy.

(4) Upgraded physical parameterizations and higher horizontal resolution substantially improve seasonal cycle of precipitation. For temporal variability at different scales (decadal, interannual, seasonal, and diurnal), model performance varies. Higher horizontal resolution does not well simulate the negative correlation between the Asian monsoon index and summer precipitation in the middle and lower reaches of the Yangtze River in China. CAM6α with 1° and 0.25° simulate the

opposite anomaly during the positive PDO phases over Southern China, two CAM5 versions with different resolutions show the similar opposite anomaly. Upgraded physical parameterizations help CAM6 to simulate the drying tendency over Southern China during El Nino years.

(5) CAM6α models generally have improvements over CAM5 in simulating diurnal cycle of precipitation in East Asia, due to the CLUBB parameterization. CLUBB has prognostic moments that provide memory to facilitate the initiation of shallow convection in the mid-morning and early afternoon.

Overall, CAM6 demonstrates better performance over CAM5, but the model fidelity at the regional scale still needs further improvement. Higher resolution improves CAM6 performance over Asia in many aspects, but some metrics are degraded. The simulated differences between 1º and 0.25º horizontal resolution are also dependent on CAM model versions. Careful validation of CAM6 performance at a regional scale is required before any quantitative statement about climate attribution and projections can be made.

**Code availability**

Model code of CESM2 is released at http://www.cesm.ucar.edu/models/cesm2/. Readers can download the current release code using the following command: git clone -b release-cesm2.1.0 https://github.com/ESCOMP/cesm.git.

**Data availability**

The model data are archived at https://doi.org/10.5281/zenodo.2548255. Details about MERRA-2 can be found at http://gmao.gsf.nasa.gov/pubs/office_notes. Details about APHRODITE can be found at http://www.chikyu.ac.jp/precip/. The observational EASMI is from http://ljp.gcess.cn/dct/page/65577. ENSO index is from http://research.jisao.washington.edu/data/cti/. The PDO index is from: http://research.jisao.washington.edu/pdo/PDO.latest. TRMM is from https://pmm.nasa.gov/data-access/downloads/trmm. The CPC dataset is from https://www.esrl.noaa.gov/psd/data/gridded/data.cpc.globaltemp.html.

**Author contributions**

WD and CW designed the study, LL led the analysis, AG performed CESM simulations, NR and SB performed CAM5-0.25º simulation. LL, YX and ZW prepared the manuscript with contributions from all co-authors.

**Competing interests**

The authors declare that they have no conflict of interest.

## Acknowledgments

We thank Richard Neale in National Center for Atmospheric Research (NCAR) for the code for computing the diurnal cycle. We also thank Patrick Callaghan and Colin Zarzycki for assistance with the original CAM6 high resolution simulations. We thank Julio Bacmeister for providing CAM5 data. This study was supported by the National Key Research and Development Program of China (2016YFA0602701) and National Natural Science Foundation of China (grant 41805053). C. Wu thanks the support of National Natural Science Foundation of China (grant 41830966). The National Center for Atmospheric Research is supported by the U. S. National Science Foundation. Portions of this study were supported by the Regional and Global Model Analysis (RGMA) component of the Earth and Environmental System Modeling Program of the U.S. Department of Energy's Office of Biological & Environmental Research (BER) Cooperative Agreement # DE-FC02-97ER62402, and the National Science Foundation. Computing resources (ark:/85065/d7wd3xhc) were provided by the Climate Simulation Laboratory at NCAR's Computational and Information Systems Laboratory, sponsored by the National Science Foundation and other agencies. An award of computer time was provided by the Innovative and Novel Computational Impact on Theory and Experiment (INCITE) program. This research used resources of the Argonne Leadership Computing Facility, which is a DOE Office of Science User Facility supported under Contract DE-AC02-06CH11357.

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

|  | CAM5-1º | CAM6α-1º | CAM5-0.25º | CAM6α-0.25º |
|---|---|---|---|---|
| Deep convection | ZM1995 | ZM1995 | ZM1995 | ZM1995 |
| Shallow convection | PB2009 | CLUBB | PB2009 | CLUBB |
| Planetary boundary layer | BP2009 | CLUBB | BP2009 | CLUBB |
| Warm cloud macrophysics | P2014 | CLUBB | P2014 | CLUBB |
| Ice cloud macrophysics | G2010 | G2010 | G2010 | G2010 |
| Cloud Microphysics | MG1 | MG2 | MG1 | MG2 |
| Aerosols | MAM3 | MAM4 | MAM3 | MAM4 |
| Horizontal resolution (latitude * longitude) | 0.9º × 1.25º | 0.9º × 1.25º | 0.23º × 0.31º | 0.23º × 0.31º |
| Timestep | 1800s | 900s | 1800s | 900s |
| The deep convective time scale | 3600s | 3600s | 3600s | 3600s |

**Table 1. Four CAM versions with major physical schemes and the time step listed in abbreviations. ZM1995 is *Zhang and McFarlane* [1995]. PB2009 is *Park and Bretherton* [2009]. BP2009 is *Bretherton and Park* [2009]. P2014 is *Park et al.* [2014]. MG1 is *Morrison and Gettelman* [2008]. MG2 is *Gettelman and Morrison* [2015]. G2010 is *Gettelman et al.* [2010]. CLUBB is *Golaz et al.* [2002a]. MAM3 is *Liu et al.* [2012]. MAM4 is *Liu et al.* [2016].**

| Differences due to physical parameterizations and high resolution | Southwestern China | | Northern China | | Maritime continent | |
|---|---|---|---|---|---|---|
| | Physic P. | High Res. With CAM6α/CAM5 | Physic P. | High Res. With CAM6α/CAM5 | Physic P. | High Res. With CAM6α/CAM5 |
| TREFHT (°C) | -0.21 | **-0.42[a]**/-0.35 | -0.01 | **0.48**/0.37 | **-0.50** | **-0.42/-0.64** |
| LHFLX (W/m²) | **2.28** | **-4.03/-1.87** | **2.24** | **-1.86**/0.14 | **5.81** | **2.01/9.17** |
| SHFLX (W/m²) | **-3.73** | **-7.37/12.71** | 0.10 | **1.73/1.09** | **6.16** | **5.13/7.16** |
| FSDSC (W/m²) | -0.48 | **-3.03/-1.14** | **-1.23** | **-3.46/-3.17** | -0.55 | -0.12/**2.96** |
| FSDS (W/m²) | **7.08** | **5.32/13.25** | **3.38** | **2.87/2.45** | **20.79** | **9.10/21.57** |
| FLDS (W/m²) | **-5.75** | **-4.00/-8.67** | 0.08 | -0.12/**1.26** | **-9.28** | **-3.24/-8.01** |
| CLDTOT (%) | **-1.46** | 0.51/**-3.88** | 0.56 | -0.68/**-2.75** | **1.70** | **2.80/-4.37** |
| INT_Q (kg/kg) | **-0.12** | -0.01/**-0.13** | -0.13 | 0.04/-0.04 | **-0.23** | -0.06/**-0.39** |
| PRECT (mm/day) | **-0.44** | **-1.06/-1.86** | -0.24 | -0.12/**-0.30** | 0.35 | **1.19/0.90** |
| PRECC (mm/day) | **-0.38** | **-0.81/-0.98** | 0.01 | **-0.31/-0.33** | **0.62** | **-1.57/-1.74** |
| PRECL (mm/day) | -0.06 | -0.25/**-0.88** | **-0.25** | 0.19/0.03 | **-0.27** | **2.76/2.64** |

**Table 2. Simulation differences due to physical parameterizations (CAM6α-1° minus CAM5-1°) and high horizontal resolution (CAM6α-0.25° minus CAM6α-1°/CAM5-0.25° minus CAM5-1°), respectively. The values are averages over Southwestern China, Northern China and Maritime continent during 1980-2004. Descriptions of variables are:**

**TREFHT**, Surface air temperature; **LHFLX**, Surface latent heat flux; **SHFLX**, Surface sensible heat flux; **FSDSC**, Clearsky downwelling solar flux at surface; **FSDS**, Downwelling solar flux at surface; **FLDS**, Downwelling longwave flux at surface; **CLDTOT**, Vertically-integrated total cloud; **INT_Q**, Vertically-integrated humidity; **PRECT**, Total precipitation rate; **PRECC**, Convective precipitation rate; **PRECL**, Large-scale precipitation rate.

5    [a] **Statistically significant differences are emphasized in bold (95% confidence level from a two-sided t test).**

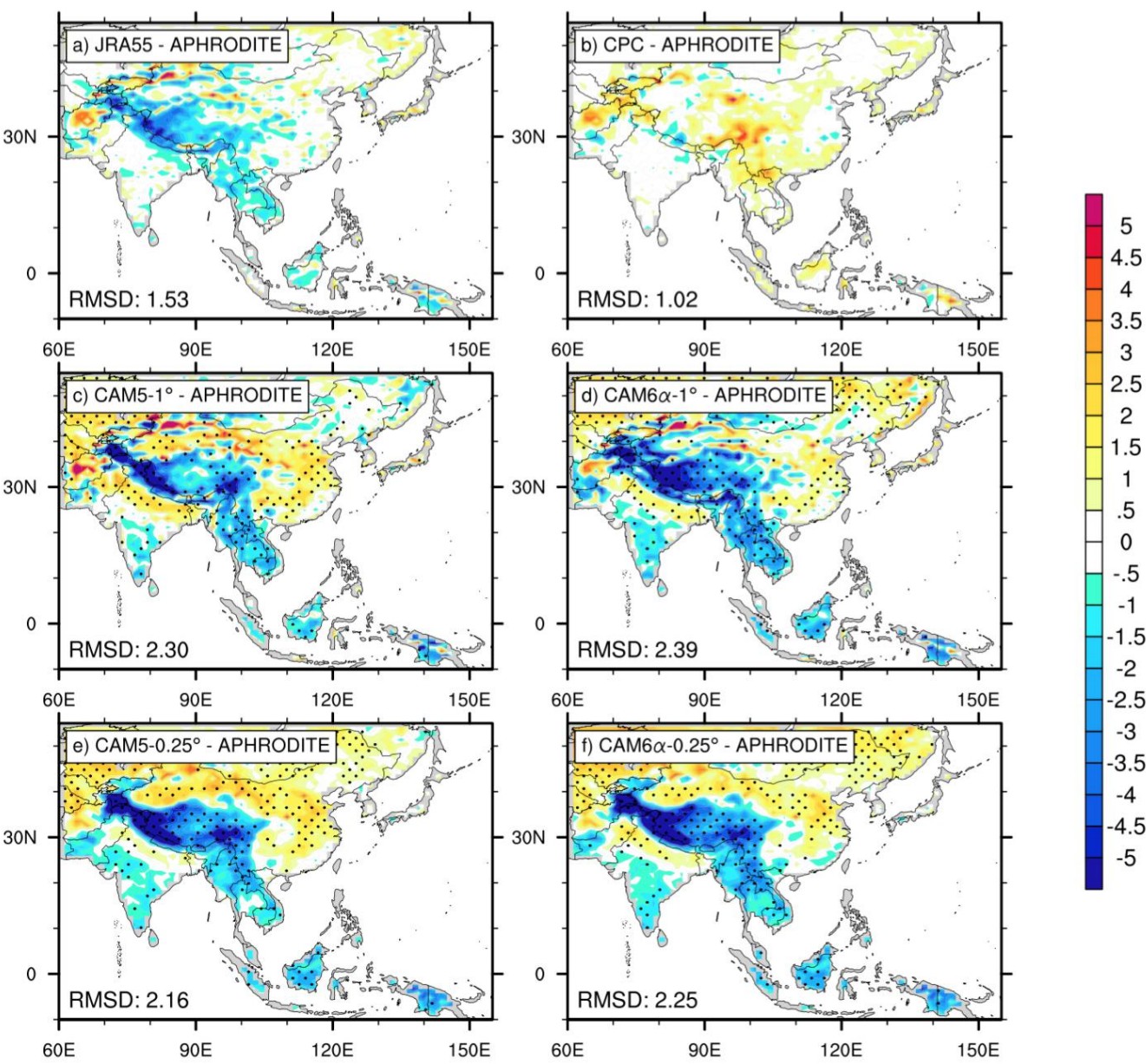

**Figure 1. The difference of 1980-2004 annual mean surface air temperature (℃) between observation (APHRODITE) and (a) JRA55, (b) CPC, (c) CAM5-1º, (d) CAM6α-1º, (e) CAM5-0.25º and (f) CAM6α-0.25º, respectively. The values at the lower left of each panel indicate the root mean squared difference (RMSD) relative to APHRODITE. Grid points in panel (c)-(f) are stippled if**

the absolute difference between the respective model and APHRODITE is larger than that between JRA55 and APHRODITE. All the data were interpolated to 1º resolution.

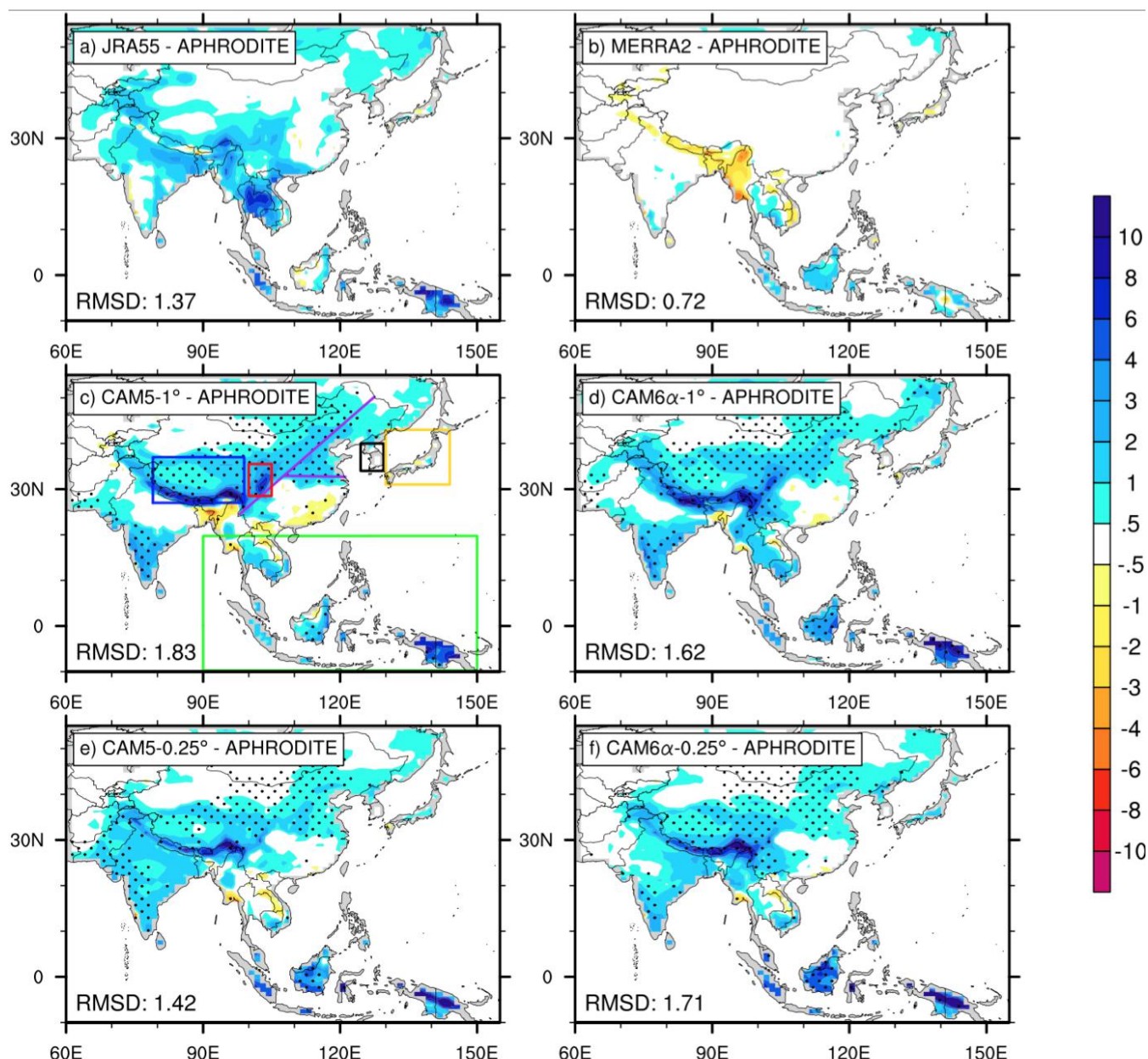

**Figure 2. The difference of 1980-2004 annual mean precipitation (mm/day) between APHRODITE and (a) JRA55, (b) MERRA2, (c) CAM5-1º, (d) CAM6α-1º, (e) CAM5-0.25º and (f) CAM6α-0.25º, respectively. The values at the lower left of each panel indicate the root mean squared difference (RMSD) relative to APHRODITE. Grid points in panel (c)-(f) are stippled when the absolute value of difference between model and APHRODITE is larger than that of between JRA55 and APHRODITE. All the data were interpolated to 1º resolution. The boxes with different color of Figure 2c are (1) Tibet (blue box): 27°N–37°N, 79°E–99°E; (2)**

Southwestern China (red box): 28.5°N–35.5°N, 100°E–105°E; (3) Korea (black box): 34°N–40°N, 124.5°E–129.5°E; (4) Japan (gold box): 31°N–43°N, 130°E–144°E; (5) the Maritime Continent (green box): 9.75°S–19.75°N, 90°E–150°E. The "India" average is entirely within mainland India. "Northern China" and "Southern China", are also defined in Figure 2c, as the where "Northern China" north of the "Qin Mountain and Huai River" at 32.8°N, and "Southern China" south of this. The western boundary of "Northern China" and "Southern China" is a straight line named as "Hu-Huanyong Line" between Heihe (50.2°N, 127.5°E) and Tengchong (24.5°N, 98.0°E).

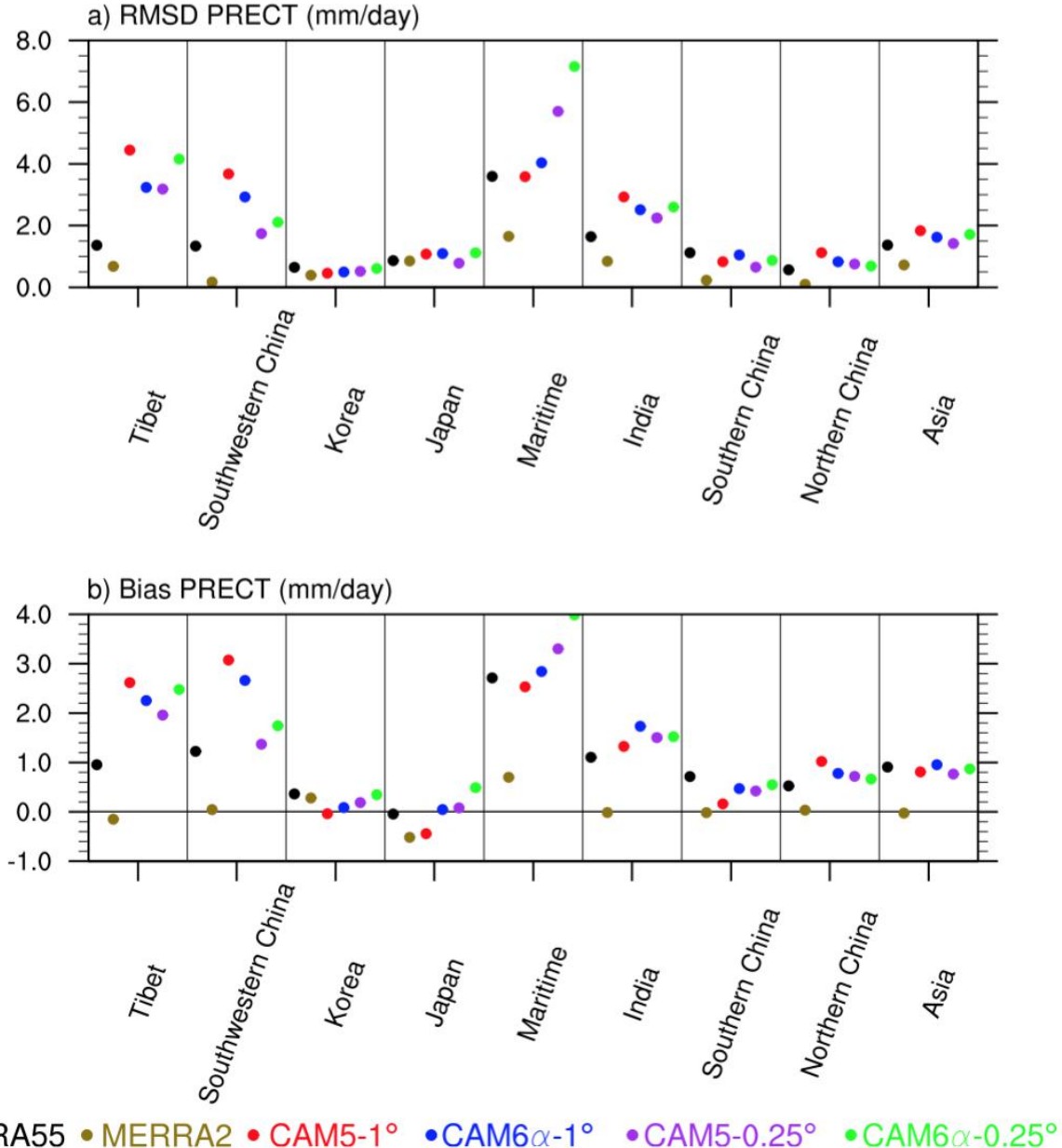

**Figure 3.** The root mean squared difference (RMSD) and bias of annual mean precipitation (PRECT, mm/day) relative to APHRODITE for JRA55, MERRA2, CAM5-1º, CAM6α-1º, CAM5-0.25º and CAM6α-0.25º over 6 regions: Tibet: 27°N–37°N, 79°E–99°E; Southwestern China: 28.5°N–35.5°N, 100°E–105°E; Korea: 34°N–40°N, 124.5°E–129.5°E; Japan: 31°N–43°N, 130°E–144°E; Maritime: 9.75°S–19.75°N, 90°E–150°E; Asia: 5°N–55°N, 60°E–140°E. Three other domains are following geographical boundary and climatic zones for India, Northern China and Southern China as in *Lin et al.* [2018]. The "India" average is the

entirely within mainland India. The western boundary of "Northern China" and "Southern China" is a straight line (shown in Figure 2c) named "Hu-Huanyong Line" between Heihe (50.2°N, 127.5°E) and Tengchong (24.5°N, 98.0°E). The separation of "Northern China" and "Southern China" (shown as the straight line in Figure 2c) is along the latitude of "Qin-Mountain and Huai-River" (32.8°N).

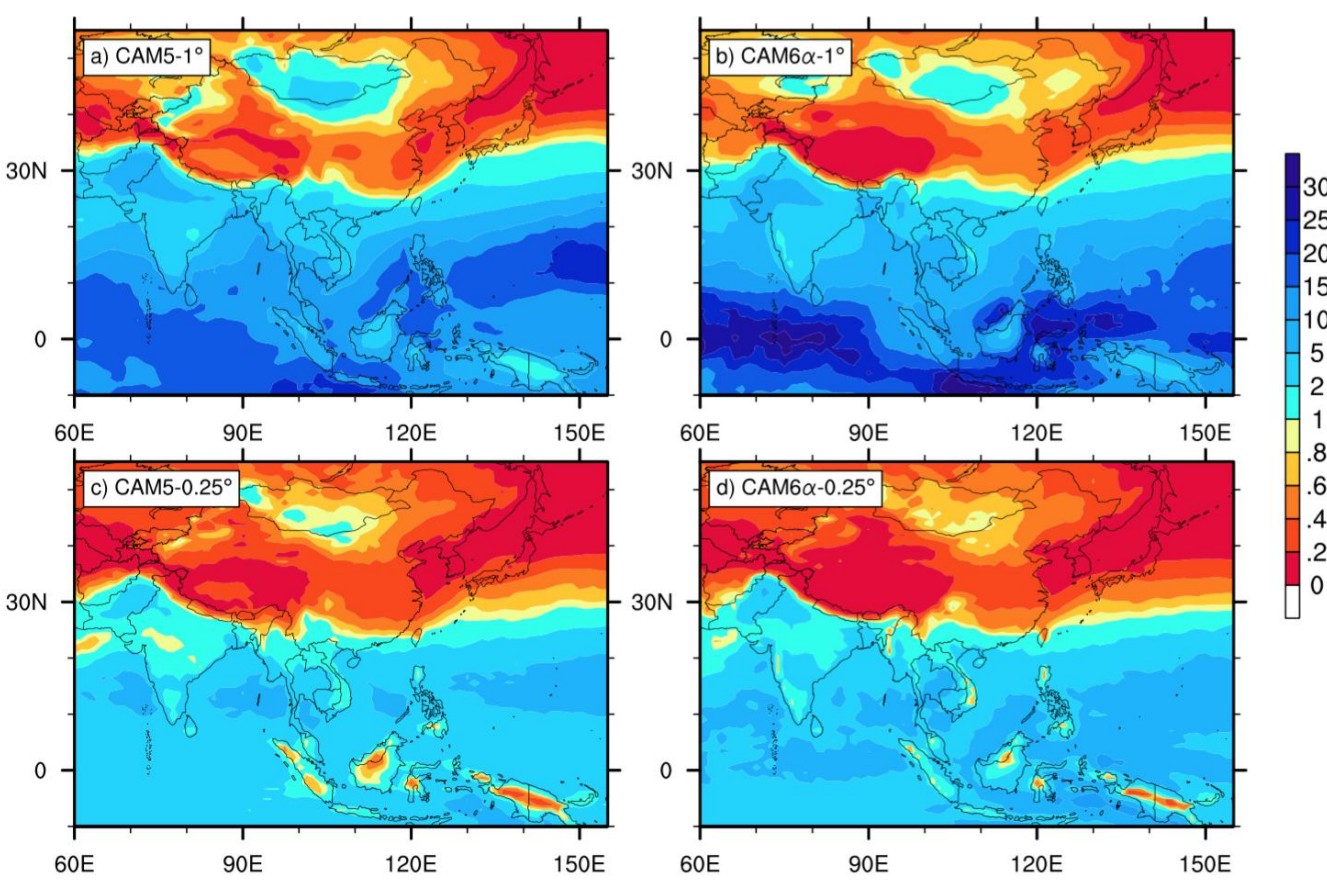

**Figure 4. The 1980-2004 convective / large-scale precipitation ratio (unitless) for the a) CAM5-1˚, b) CAM6α-1˚, c) CAM5-0.25˚ and d) CAM6α-0.25˚, respectively.**

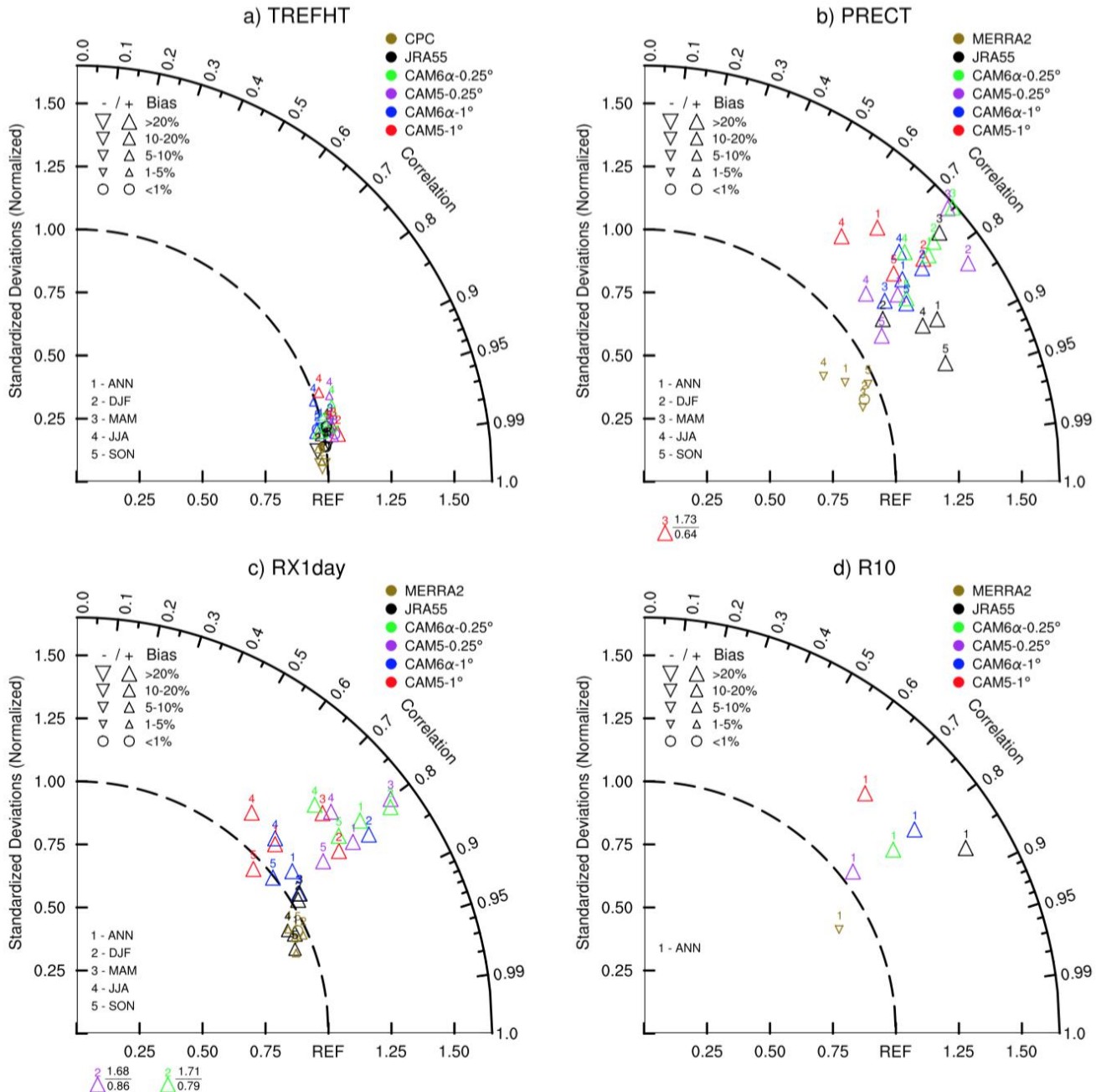

**Figure 5. Taylor diagrams for (a) surface air temperature, (b) precipitation, (c) RX1day (the maximum daily precipitation during each month of a year, units: mm/day), (d) R10 (number of days with precipitation more than 10 mm, units: days). These two extreme indices are suggested by the Expert Team for Climate Change Detection and Indices [Zhang et al., 2011]. Note that R10, by design, only has annual mean, but not seasonal mean. Spatial correlations and normalized RMS (root mean square) are**

calculated for ANN (with 1 on the top), DJF (2, winter), MAM (3, spring), JJA (4, summer), SON (5, fall). APHRODITE is used as the benchmark over Asia for 1980-2004. All the data were interpolated to 1º resolution.

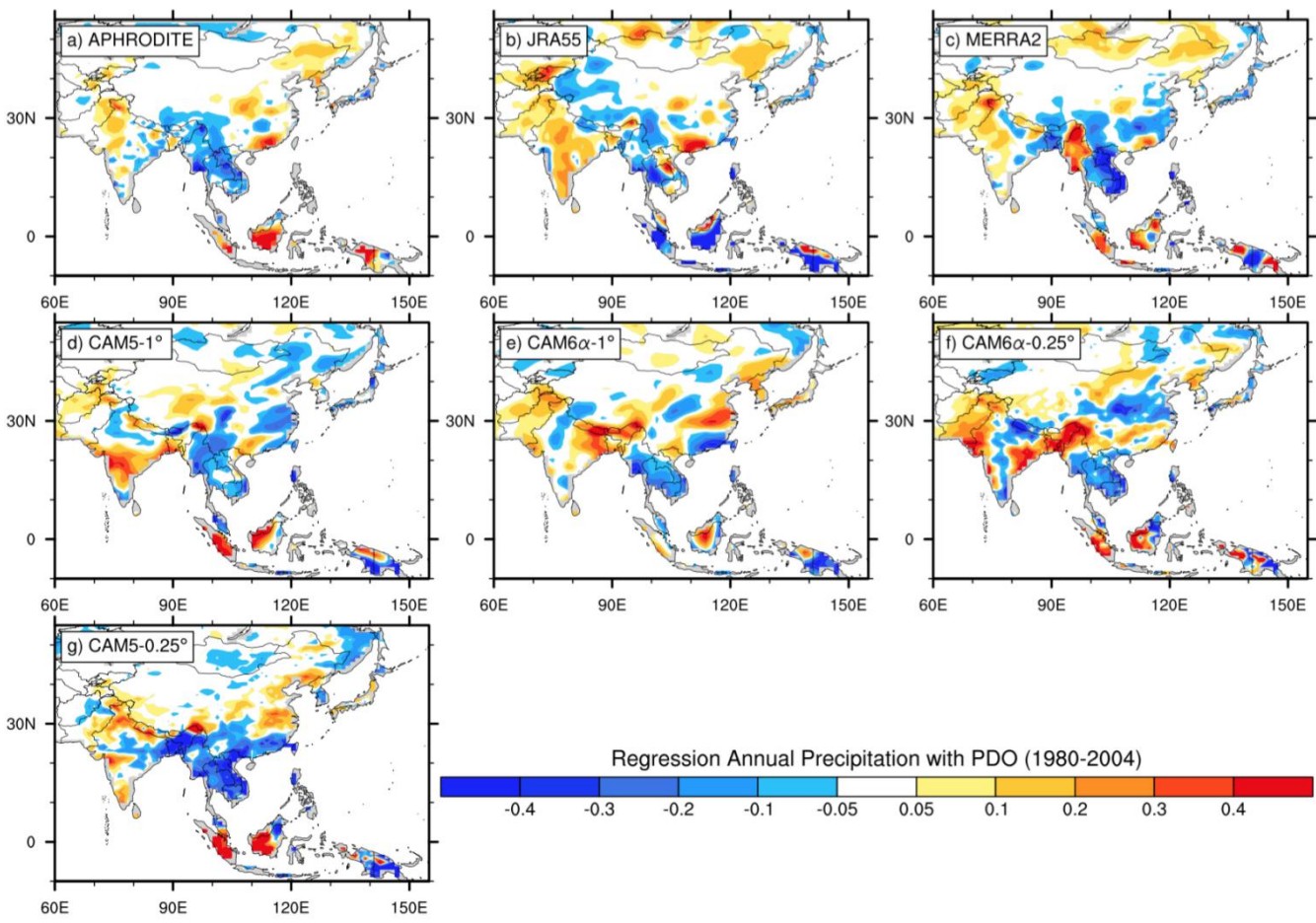

5   **Figure 6. Regression of annual mean precipitation (mm/day) onto observed PDO indices for 1980-2004.**

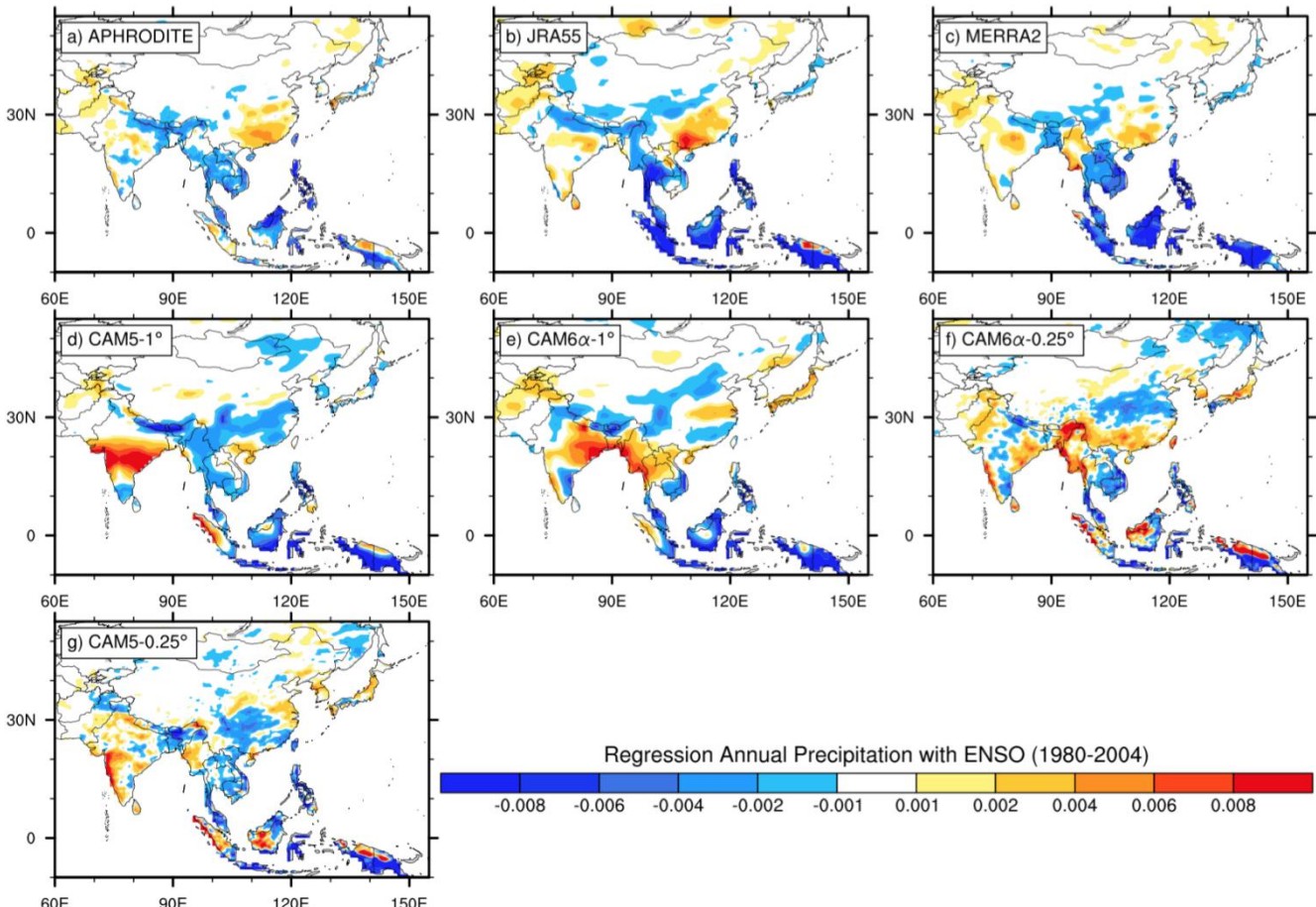

Regression Annual Precipitation with ENSO (1980-2004)

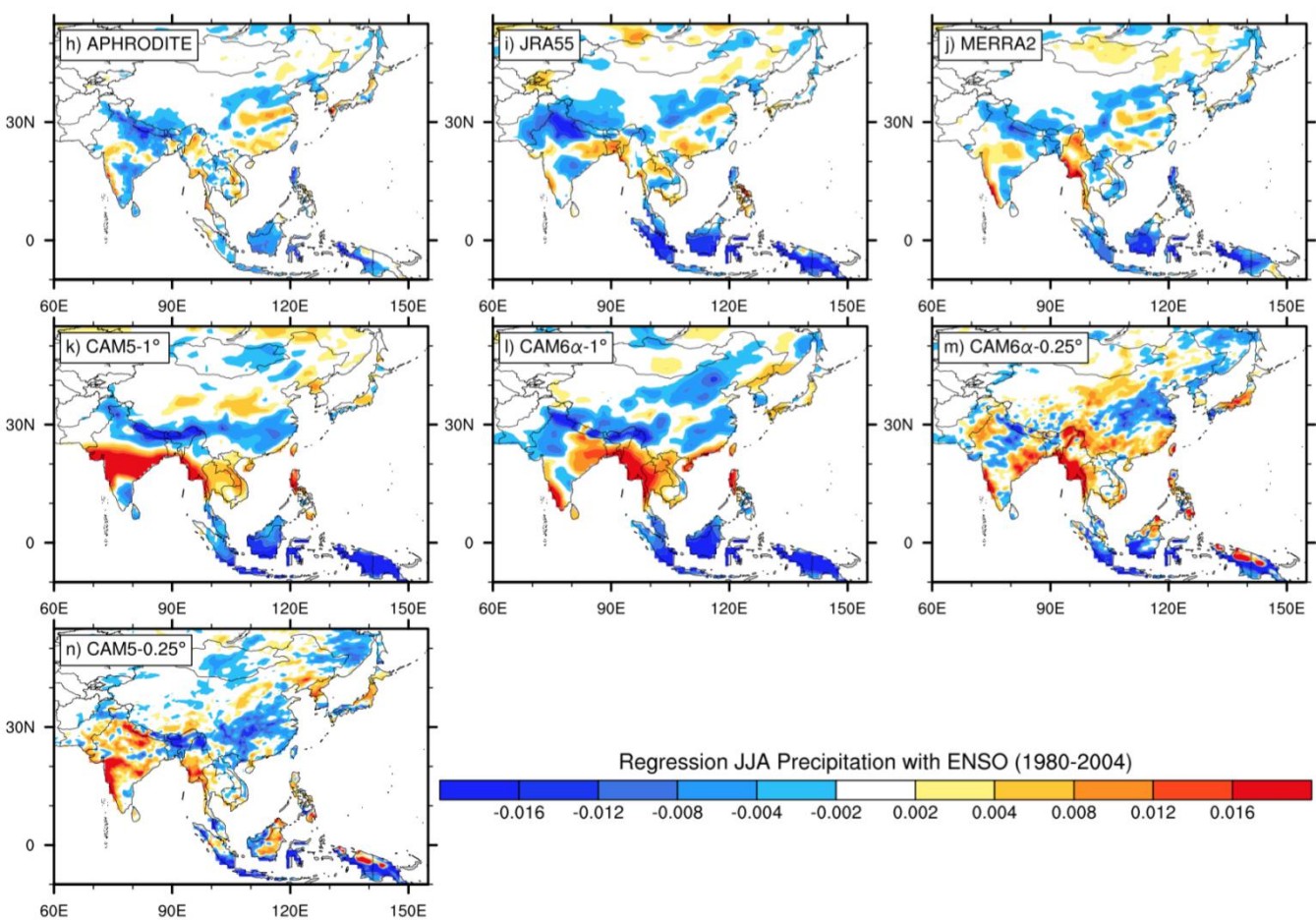

**Figure 7. Regression of annual mean (a-g) and summer (JJA, h-n) precipitation onto the observed ENSO index (mm/day) for 1980-2004. Top color bar is for panel a-g, bottom color bar is for panel h-n.**

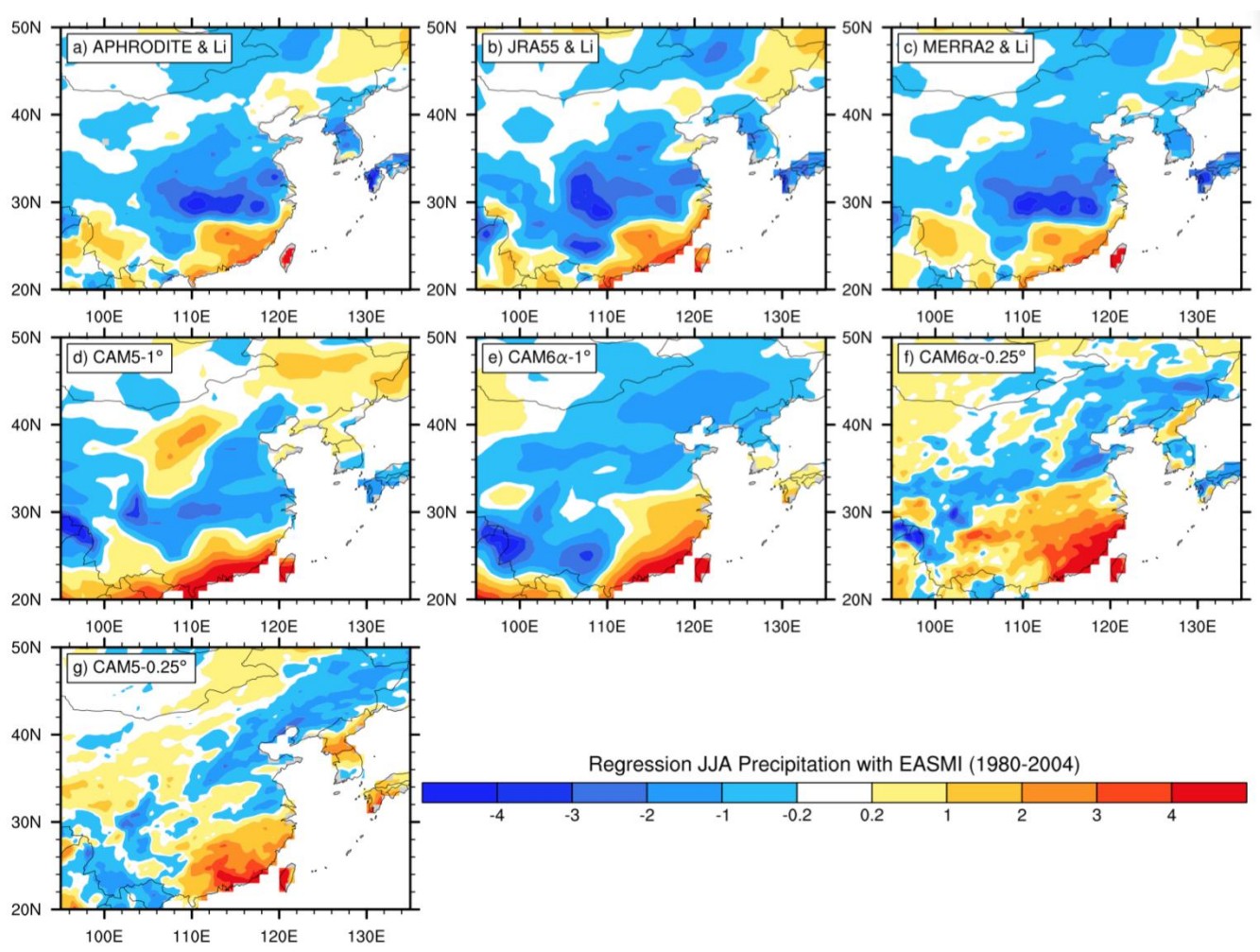

**Figure 8. Regression of summer (JJA) precipitation onto EASMI (mm/day) for 1980-2004.**

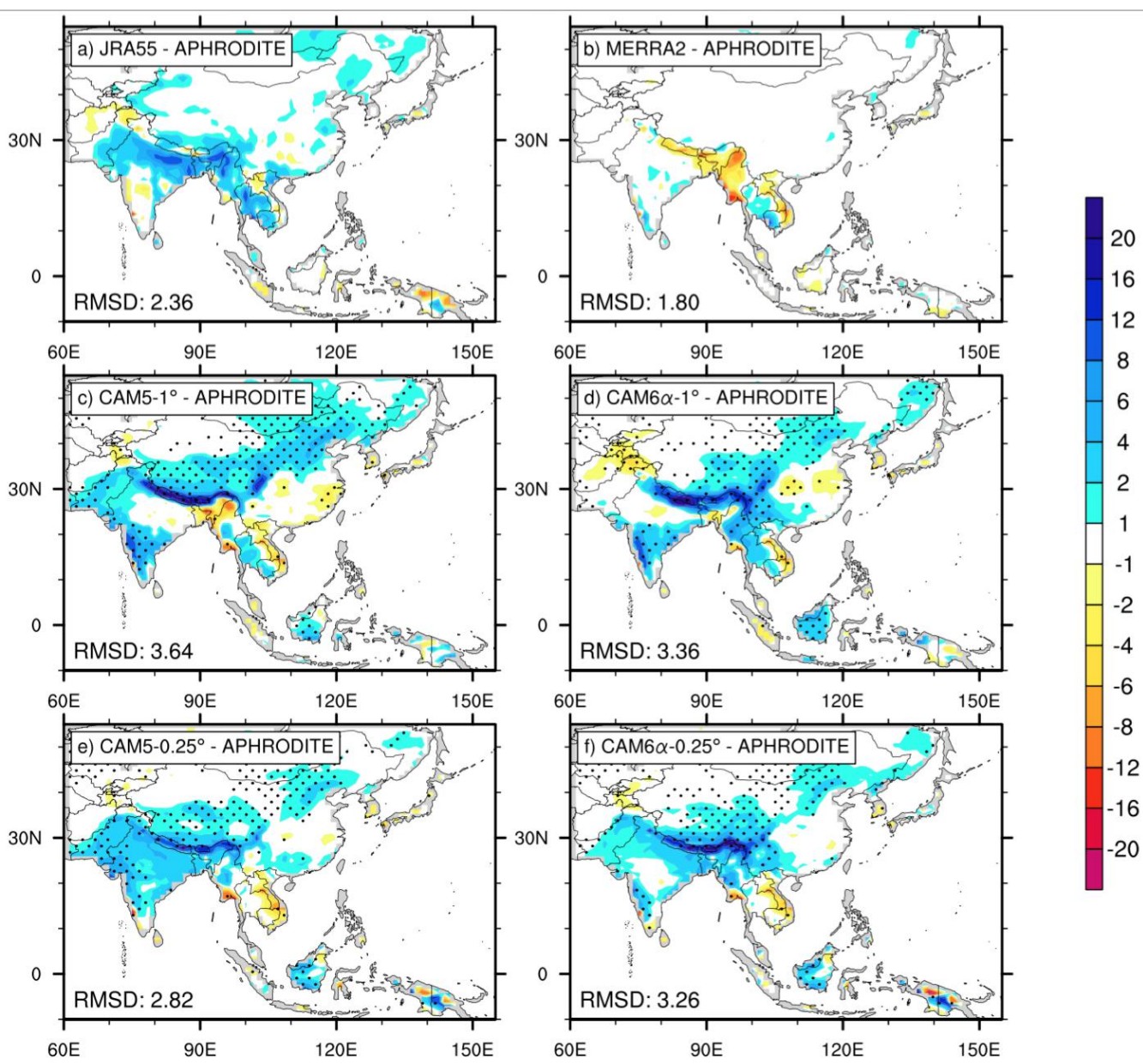

**Figure 9. The difference of summer minus winter (JJA - DJF) precipitation (mm/day) between APHRODITE and a) JRA55, b) MERRA2, c) CAM5-1°, d) CAM6α-1°, e) CAM5-0.25° and f) CAM6α-0.25°, respectively. The values at the lower left of every panel indicate the root mean squared difference (RMSD) relative to APHRODITE. Grid points in panel (c)-(f) are stippled when the absolute value of difference between model and APHRODITE is larger than that between JRA55 and APHRODITE.**

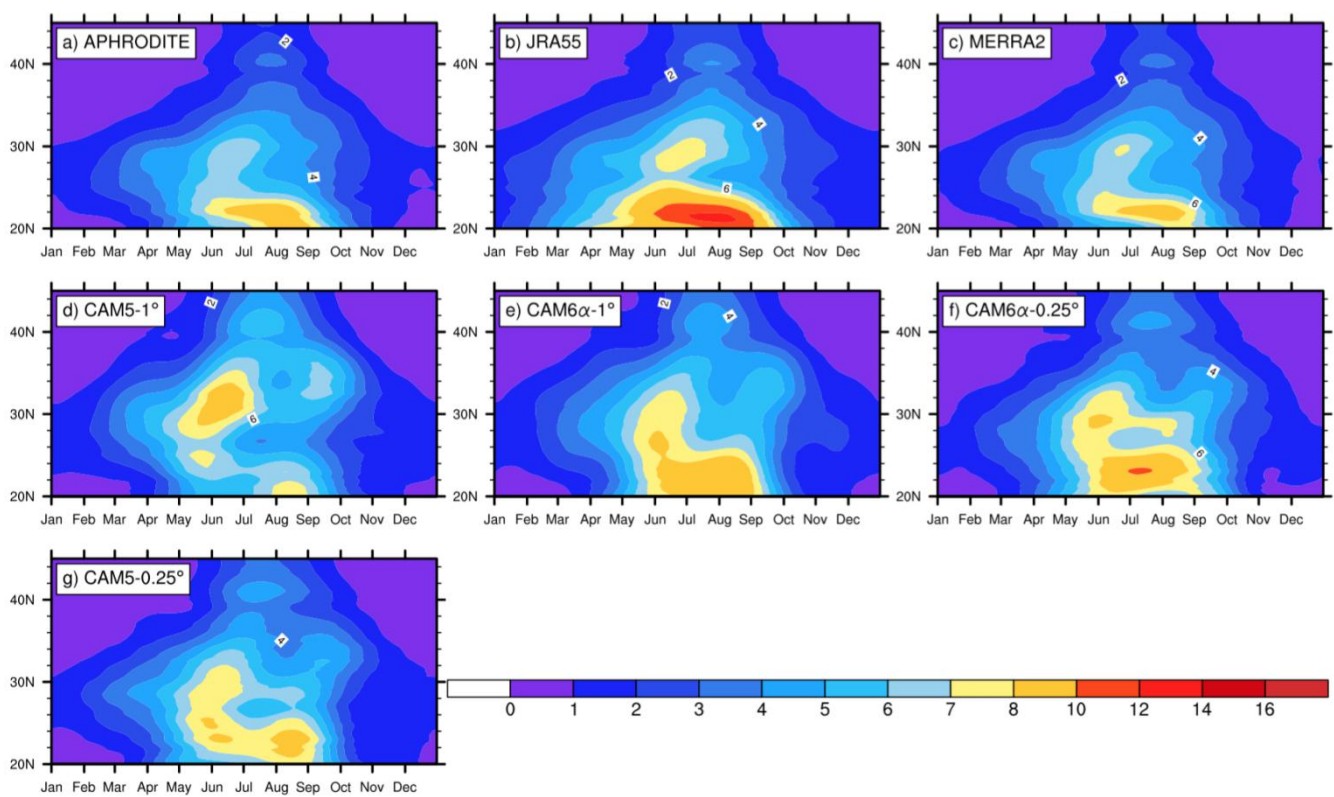

**Figure 10. Annual cycle of regional mean precipitation rate (mm/day) for 1980-2004 within Part of Asia (between 100˚E and 125˚E): a) APHRODITE, b) JRA55, c) MERRA2, d) CAM5-1˚, e) CAM6α-1˚, f) CAM6α-0.25˚ and g) CAM5-0.25˚.**

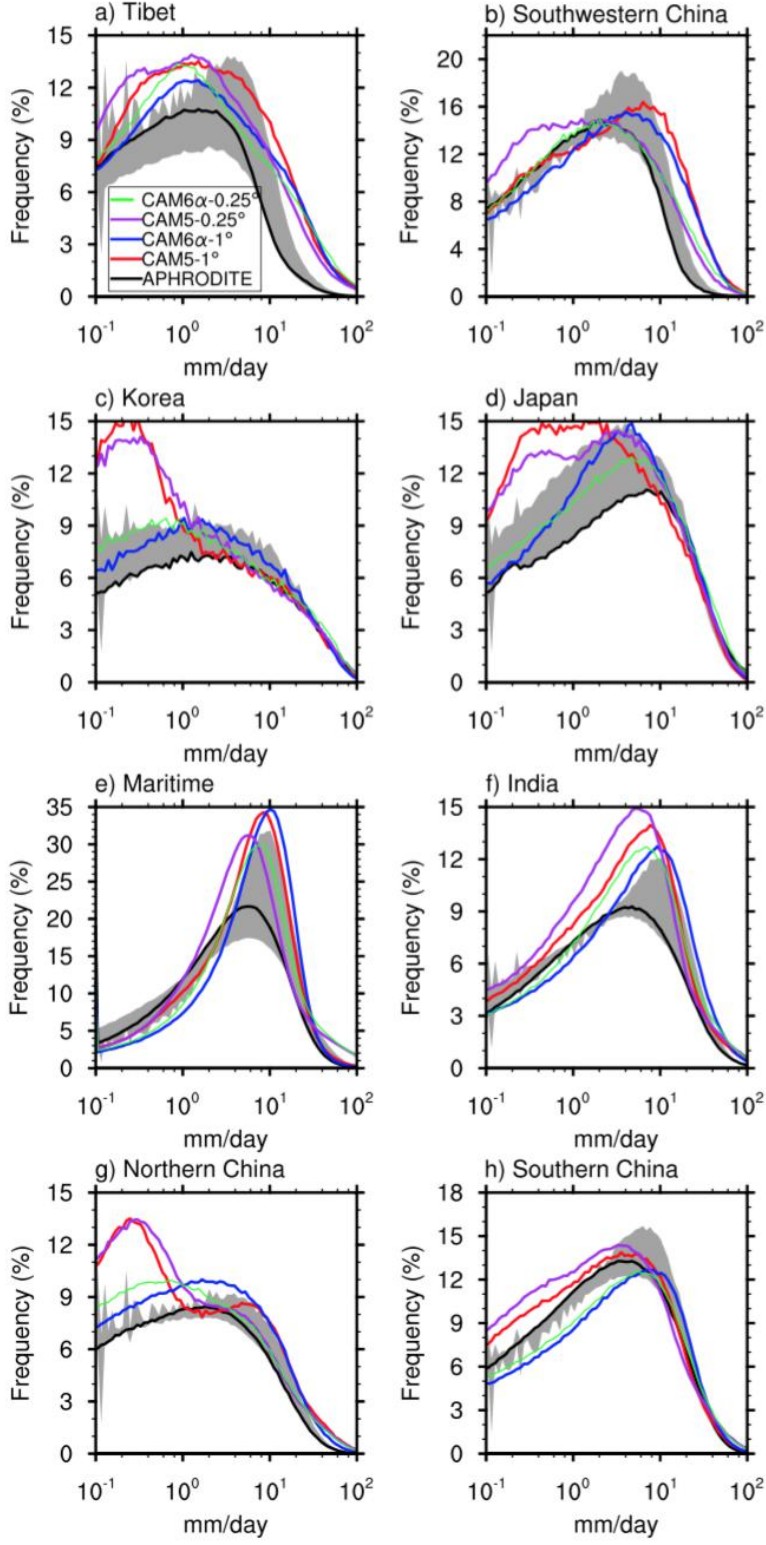

**Figure 11. Frequency distribution of daily precipitation (mm/day) over a) Tibet, b) Southwestern China, c) Korea, d) Japan, e) maritime continent (see the boxes in Figure 2c), f) India, g) Northern China and h) Southern China are for 1980-2004. Black lines (shading) for the APHRODITE (the maximum and minimum of APHRODITE, JRA55 and MERRA2). Red solid lines are for CAM5-1º; blue solid lines for CAM6α-1º; green solid lines for CAM6α-0.25º; purple solid lines for CAM5-0.25º. All data were interpolated to 1º resolution before regional average.**

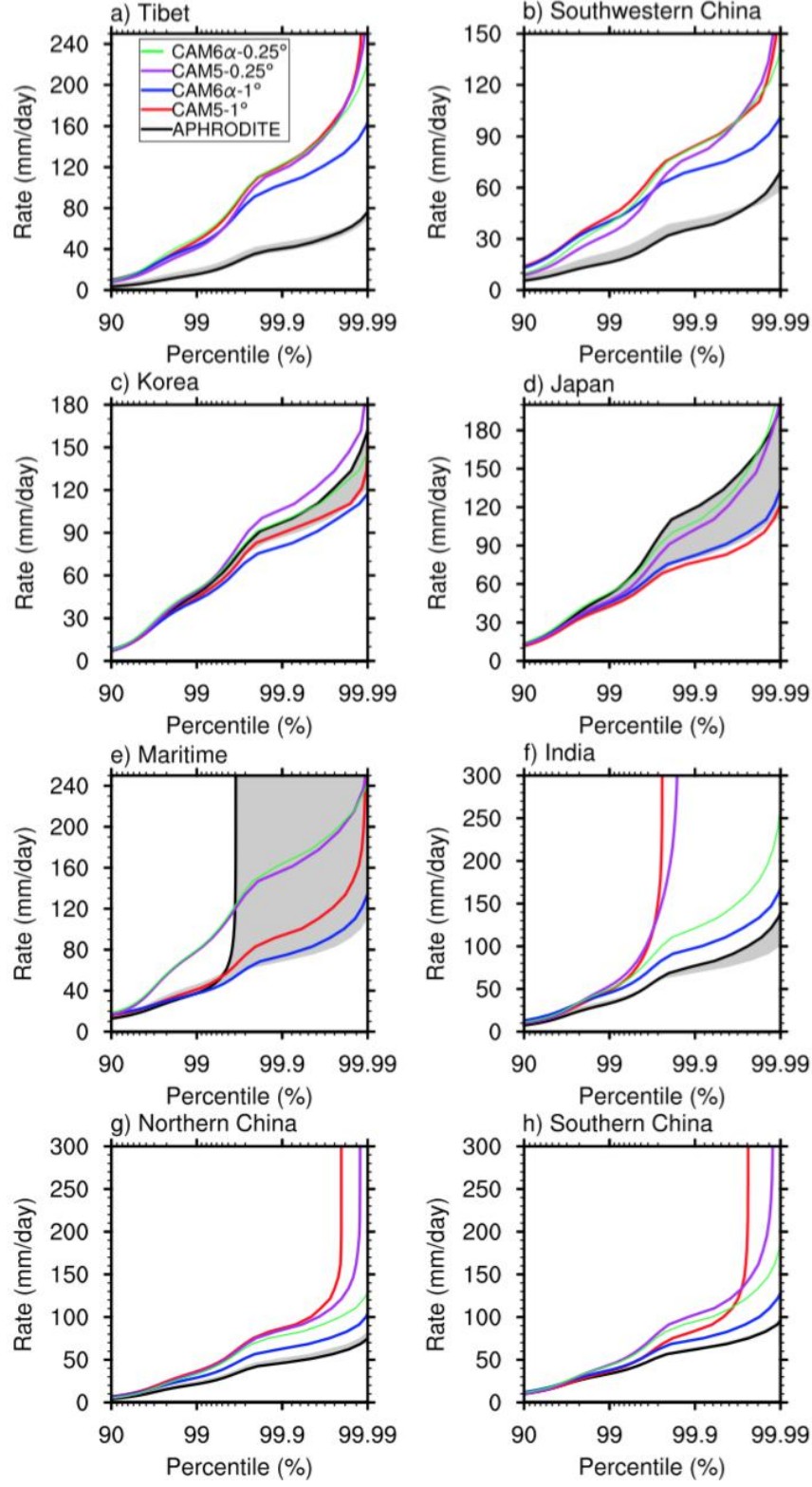

**Figure 12. Similar to Figure 11, but for daily precipitation rates (mm/day) as a function of percentile.**

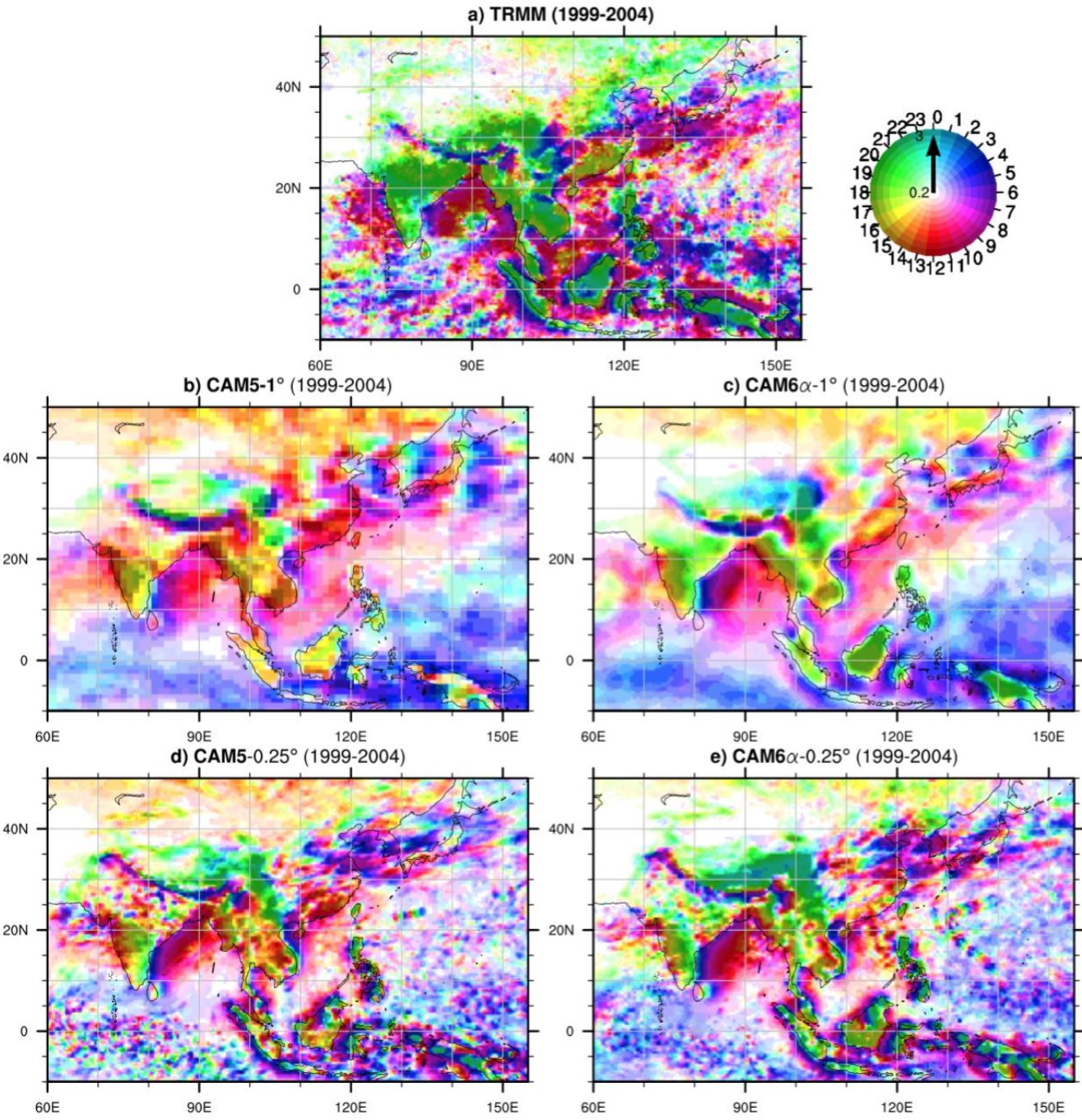

Figure 13. Diurnal cycle of precipitation in June (6-year average from 1999-2004) of a) TRMM, b) CAM5-1º, c) CAM6α-1º, d) CAM5-0.25˚ and e) CAM6α-0.25º. The local time peak of the diurnal cycle is shown in color on the color wheel. The intensity of the color is the amplitude of precipitation.

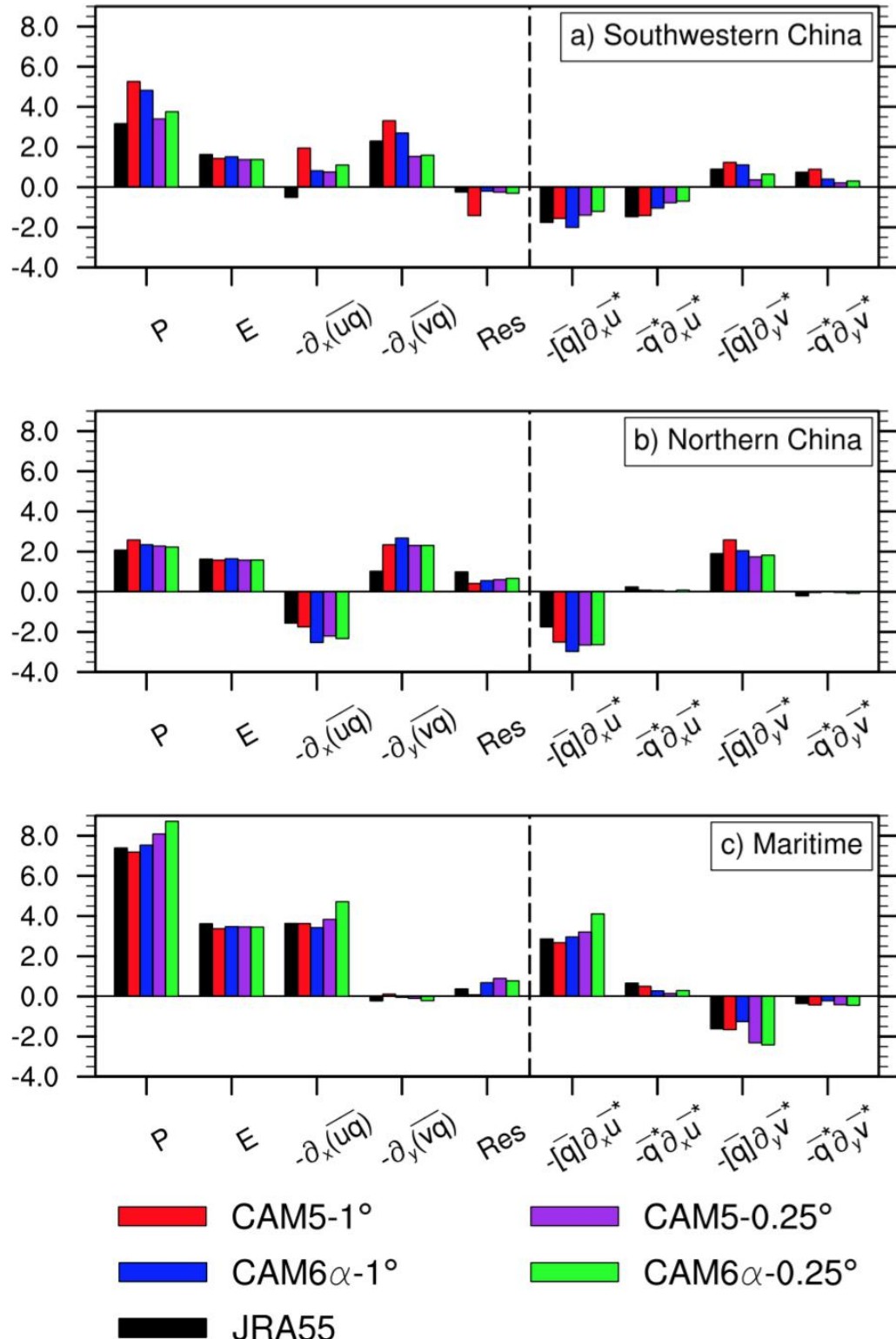

**Figure 14. Moisture budget over a) Southwestern China, b) Northern China and c) Maritime continent (see the boxes of Figure 2c) for 1980-2004, including precipitation, evaporation, zonal and meridional moisture flux convergence, residual and (right to dashed line) major processes contributing to moisture flux convergence.**