# Peer review of "CAM6 simulation of mean and extreme precipitation over Asia: Sensitivity to upgraded physical parameterizations and higher horizontal resolution"

_Geoscientific Model Development, 2019_

## Referee Comment (RC1) · Anonymous Referee #1 · 21 Feb 2019

This paper documents the precipitation over Asia for CAM5 and two resolution configurations of a quasi-CAM6 version. The authors find that many aspects of the simulated precipitation features (both mean state and variability) are improved with CAM6, though persistent biases remain. Further, higher resolution CAM6 can further improve on the low resolution CAM6 results.

Overall, I found this to be a fairly well written paper that was concise and contained lots of interesting results. My main critique of this paper is not so much with what the authors show but with what the authors DIDN'T show. I recommend publication after

the authors have considered addressing my two major points of critique and improve the clarity of the paper by addressing the two minor points.

Major Points

1) The authors present CAM5-1 degree, while presenting CAM6-1 degree and CAM6-0.25 degree. Why not show results from CAM5-0.25 degree? I feel neglecting the inclusion of this configuration leaves a hole in the story the authors are trying to tell. It would be very interesting to see how two very different versions of CAM respond to increases in resolution and whether the improvements seen in CAM6 high resolution are, in fact, unique to CAM6. 2) Page 6, line 2, the authors state "while the poor performance of CAM6, especially over Martime continent will be dealt with in a separate paper". Why? This would be the appropriate paper to discuss this topic and the sudden neglection/omission of this topic gave the paper a rather disjointed feel since I felt like a crucial piece of the story was missing.

Minor Points

1) The authors need to be more explicit about what they mean when they refer to regions such as "Sichuan" or "southern China". I know figure 2c has boxes denoting regions, yet these boxes are not labeled anywhere in the figure or caption. Further, these regions are referred to in Figure 1 and it required efforts of my own to try to identify what the authors were referring to in reference to "Sichuan" etc. Explicitly defining these regions EARLY in the paper will go a long way towards improving the clarity of the paper. 2) Table one should include any difference in the time step between the 1 degree and 0.25 degree model. In addition, was the deep convective time scale adjusted in the 0.25 degree simulation versus the 1 degree simulation?
* * *

---

## Referee Comment (RC2) · Anonymous Referee #2 · 21 Mar 2019

**GENERAL COMMENTS**

This is an interesting paper evaluating the performance of the CAM6 prototype in the Asian region at two horizontal resolutions and comparing against CAM5. The paper is reasonably well-written and the figures are clear. I have some concerns regarding the use of reanalyses data as a second benchmark, and it would helpful to clarify how some of the resolution comparisons have been made.

Finally, it would be helpful if the manuscript could be checked carefully by a native

**English speaker to remove numerous errors.**

**SPECIFIC COMMENTS**

page 2. line 25: Actually, GA6 is not the latest atmosphere UK Met Office. al. model from the Williams et (2017; https://agupubs.onlinelibrary.wiley.com/doi/full/10.1002/2017MS001115) state that the GC3 coupled configuration, which has GA7 as its basis, will form the basis for their CMIP6 submissions. There are substantial changes between GA6 and GA7. You can still quote the studies relating to GA6, but perhaps reword this paragraph.

page 3, line 1: Enhanced model resolution has not always been demonstrated as a means to reduced model biases, especially not in the tropics. For example, Johnson et al (2016; https://link.springer.com/article/10.1007%2Fs00382-015-2614-1) showed that increasing horizontal resolution was not a solution to the South Asian monsoon biases in the Met Office GA3 model and also stated, based on past studies, that "it is difficult to attribute the monsoon improvement to any particular physics or resolution change in the atmosphere or ocean components."

page 4, line 2: "time-varying observed sea surface temperatures and sea ice" - what is the time resolution of your forcing dataset? Is it monthly interpolated or actual daily mean values?

Section 2.2: I am a little concerned that there is really only one observational rainfall dataset employed here. Rainfall from reanalyses is very dependent on the model physics, and is therefore not really a suitable benchmark. I realise that you may be constrained by the time period and horizontal resolution of your simulations, but it would be good to include more caveats on the APHRODITE data, particularly the potential lack of gauge observations in mountainous areas: how reliable are the values over Tibet, the Himalayas and the Maritime Continent islands? While I would not expect a detailed comparison between observational datasets in your study, some additional discussion on this aspect is warranted, instead of simply says in lines 18-20 of page 4 that you
use the reanalyses as a benchmark, thereby implying that this is suitable.

Similarly, there other global for surface are datasets air CPC: temperature from in situ (such measurements as https://www.esrl.noaa.gov/psd/data/gridded/data.cpc.globaltemp.html) that could be used as a second benchmark.

Figures 1 and 2: Please confirm in the caption that you have interpolated the observations/reanalyses to the model's 1 degree grid, and the APHRODITE data to the JRA-55's 0.56 degree resolution, for panels (a) to (c) of these figures?

page 5, line 17: "...to fully capture the larger uncertainty of observational datasets" - see previous comment regarding reanalyses. I suggest "...as an estimate of the large uncertainty..." would be more appropriate.

page 6, lines 13-14 and 23-24: An increase in horizontal resolution between 1 degree and 0.25 degrees is grossly insufficient to avoid the use of parameterization of clouds and convection by resolving those processes. Even in a prognostic scheme such as CLUBB, the microphysical processes are still parameterized: a prognostic increment to the condensate and rainfall is calculated by making assumptions about (i.e. parameterizing) the relationship between the thermodynamic variables and the microphysical process. Increasing resolution could change the local circulations, the thermodynamic variables and their distribution, and may make them more realistic if the processes driving those are resolved better, but this is not actually resolving the precipitation process itself better. Ultimately, as you note in line 17, the partition between the large-scale and convective rainfall in any model (including reanalyses), and how this changes with resolution, will depend on the parameterization schemes employed.

There are also several statements in this section that imply that convection parameterizations are only there to mop up instability. I do not think that this is true. They should be representing the effects of sub-gridscale convective processes (as opposed to subgridscale stratiform cloud processes, which are handled by the large-scale cloud and GMDD
precipitation parameterization) on the environment in the gridbox.

Please consider rewording these misleading sentences.

page 6, lines 16-21 and Figure 4: Despite your caveat that JRA-55 is providing a partition that is model-dependent, its inclusion in Figure 4 implies that you are considering it as a benchmark for the model comparison. I recommend that you remove this and only consider the comparison between CAM versions.

Instead, you could include in Figure 4 some evidence that the change in timestep affects the partitioning in the way that you assert in lines 26-31 (by showing results from CAM6-1 with a 10 minute timestep, perhaps?).

page 7, lines 1-13, and Figure 5: Please confirm that you have compared all of these on the same 1 degree grid resolution? This is particularly important for the RX1day and R10 statistics that are measured using threshold values against the intensity distribution, which itself will depend on the resolution of the data (even for the observations).

page 8, line 4: "higher frequency fluctuations" - are these really higher frequency fluctuations? It is still an interannual variation, albeit of seasonal mean values.

Also, given the known influences of ENSO on boreal summer monsoon rainfall over Asia, it would be interesting to compare the regression of JJA mean rainfall against ENSO in Figure 7, as well as that of the annual mean rainfall.

Section 4.3 and Figures 11 and 12: Similar to my previous comment regarding Figure 5, please confirm that everything has been reduced to the same horizontal resolution before doing this analysis.

Further, you have suddenly taken the mean of the observations and reanalyses here, in what is perhaps the hardest test for the models. It would be advisable to state or show how these distributions vary between APHRODITE and the two reanalyses - is this really captured with the one standard deviation? There are only three datasets, so why not show the envelope as shading instead (and APHRODITE as the solid line)?
page 9 line 16: The CAM6-0.25 also looks worse for heavy rainfall (than CAM6-1) for the Sichuan and Tibet regions.

page 10, lines 3-5: "High-resolution simulations..." - where is your evidence for this statement? Even if it is true (of which I am not sure), there is also horizontal moisture advection which will be different at higher resolution.

page 10, line 6: Presumably the increased downward solar radiation in CAM6 is related to the improved diurnal cycle? Please state this, if it is the case.

Section 5: Overall, I do not find this section convincing, nor does it add much to the findings of the study. You make several statements about the impact of the new physics in CAM6 on the balance of processes and the large-scale/convective rainfall partition that are speculative and not supported by evidence. I would suggest that you remove this section and Figure 14 (and the associated bullet point (4) on page 11).

page 11 line 17: "...better performance over Sichuan basin..." - is this true? It looks worse than CAM6 at 1 deg in Figure 12.

**TECHNICAL CORRECTIONS**

[Note that, in addition to the points raised below, there are numerous wording and grammar issues that require careful editing by a native English speaker]

page 2 line 11: "regarding to" -> relating to

page 3, line 27: Please expand CLUBB, or at least mention which part of the model physics this relates to.

page 6, line 27: "...whenever large scale condensation that process removes all liquid..." - I do not understand this sentence.

page 8, line 16: "...edge of the positive correlation is less more northward..." - more, I think.
Figure 10 caption: This is not just within Southern China.

page 9, line 1: "which is missing the persistent in CAM5-1 (the jumping cliff over July to August..." ??

Table 2 - you have not referred to this table anywhere in the manuscript. It may not be needed if you remove section 5.

---

## Author Response (AR1)

This paper documents the precipitation over Asia for CAM5 and two resolution configurations of a quasi-CAM6 version. The authors find that many aspects of the simulated precipitation features (both mean state and variability) are improved with CAM6, though persistent biases remain. Further, higher resolution CAM6 can further improve on the low resolution CAM6 results.

Overall, I found this to be a fairly well written paper that was concise and contained lots of interesting results. My main critique of this paper is not so much with what the authors show but with what the authors DIDN'T show. I recommend publication after the authors have considered addressing my two major points of critique and improve the clarity of the paper by addressing the two minor points.

Response:
We appreciate the constructive comments from the reviewer. We have taken all of the reviewer's comments into consideration and revised the manuscript accordingly. Our detailed responses are as follows.

Major Points

1) The authors present CAM5-1 degree, while presenting CAM6-1 degree and CAM6-0.25 degree. Why not show results from CAM5-0.25 degree? I feel neglecting the inclusion of this configuration leaves a hole in the story the authors are trying to tell.

It would be very interesting to see how two very different versions of CAM respond to increases in resolution and whether the improvements seen in CAM6 high resolution are, in fact, unique to CAM6.

Response:
        Thank you for raising this issue. We totally agree with you. We have complemented the CAM5-0.25 degree results into Figure 3, 11, 12, 14, and Table 2.  Basically, the response to increase in horizontal resolution depends on CAM version. The statements in below have been added into the manuscript:
        "Both CAM versions with higher resolution simulate the climatological precipitation over Northern China better. Increasing model resolution decrease the RMSD and bias of CAM5 over Tibet and Southwestern China while increase those of CAM6 (Figure 3)."
        "CAM5-0.25° overestimates the frequency of light precipitation (0.1-10 mm/day) over Southwestern China (purple line in Figure 11b)."
        "No significant differences are found between CAM5 and CAM6 over Tibet, Southwestern China, Japan and the Maritime Continent."
        "Higher resolutions in CAM6 and CAM5 both decrease the surface latent heat flux and convective precipitation over Southwestern China (Table 2). However, higher resolution leads to an opposite change in surface sensible heat flux between CAM5 and CAM6α (by 12.7 and -7.4 W/m2, respectively)."

"CAM5-0.25˚ simulates similar moisture budget with CAM6-0.25˚, while the corresponding results are different between CAM5-1˚ and CAM6-1˚ over Southwestern China and Northern China (Figure 14a and 14b). "

"With a prognostic treatment of large-scale instability and the convective response, higher horizontal resolution in CAM6 leads to better performance on the frequency distributions of daily precipitation over Southwestern China (close to the edge of Tibet Plateau) and heavy precipitation over Northern China. The improvement, however, is dependent on CAM versions."

"The simulated differences between 1º and 0.25º horizontal resolution are also dependent on CAM model versions."

We also cited two previous studies that compared the results from CAM5 versions with various resolutions with those from CAM6-0.25ºand then compared them with our results in this section:

"*Wehner et al*. [2014] found that the extreme precipitation amounts are larger as the resolution increase in CAM5."

"With a fully coupled Community Climate System Model Version 4 (CCSM4), an earlier version of CAM (CAM4), *Shields et al*. [2016] have shown that higher horizontal resolution tends to decrease convective precipitation and increase large-scale precipitation."

Wehner, M. F., Reed, K. A., Li, F., Bacmeister, J., Chen, C. T., Paciorek, C., ... & Jablonowski, C. (2014). The effect of horizontal resolution on simulation quality in the Community Atmospheric Model, CAM5. 1. *Journal of Advances in Modeling Earth Systems*, *6*(4), 980-997.
Shields, C. A., Kiehl, J. T., & Meehl, G. A. (2016). Future changes in regional precipitation simulated by a half‐degree coupled climate model: Sensitivity to horizontal resolution. *Journal of Advances in Modeling Earth Systems*, *8*(2), 863-884.

2) Page 6, line 2, the authors state "while the poor performance of CAM6, especially over Martime continent will be dealt with in a separate paper". Why? This would be the appropriate paper to discuss this topic and the sudden neglection/omission of this topic gave the paper a rather disjointed feel since I felt like a crucial piece of the story was missing.

Response:
Accepted. We have added the corresponding analysis. The specific changes are the following:

"We will explore the details that might lead to the progressive improvement over Sichuan and Northern China in Section 5, while the poor performance of CAM6, especially over Maritime continent will be dealt with in a separate paper."
==>
"We will explore the details that might lead to the progressive improvement over Southwestern China and Northern China and the poor performance of CAM6-0.25˚ over Maritime continent in Section 5."

| Differences due to physical parameterizations and high resolution | Southwestern China | | Northern China | | Maritime continent | |
|---|---|---|---|---|---|---|
| | Physic P. | High Res. With CAM6α/CAM5 | Physic P. | High Res. | Physic P. | High Res. With CAM6α/CAM5 |
| TREFHT (℃) | -0.21 | **-0.42[a]/**-0.35 | -0.01 | **0.48**/0.37 | **-0.50** | **-0.42/-0.64** |
| LHFLX (W/m²) | **2.28** | **-4.03/-1.87** | **2.24** | **-1.86**/0.14 | **5.81** | **2.01/9.17** |
| SHFLX (W/m²) | **-3.73** | **-7.37/12.71** | 0.10 | **1.73/1.09** | **6.16** | **5.13/7.16** |
| FSDSC (W/m²) | -0.48 | **-3.03/-1.14** | **-1.23** | **-3.46/-3.17** | -0.55 | -0.12/**2.96** |
| FSDS (W/m²) | **7.08** | **5.32/13.25** | **3.38** | **2.87/2.45** | **20.79** | **9.10/21.57** |
| FLDS (W/m²) | **-5.75** | **-4.00/-8.67** | 0.08 | -0.12/**1.26** | **-9.28** | **-3.24/-8.01** |
| CLDTOT (%) | **-1.46** | 0.51/**-3.88** | 0.56 | -0.68/**-2.75** | **1.70** | **2.80/-4.37** |
| INT_Q (kg/kg) | **-0.12** | -0.01/**-0.13** | -0.13 | 0.04/-0.04 | **-0.23** | -0.06/**-0.39** |
| PRECT (mm/day) | **-0.44** | **-1.06/-1.86** | -0.24 | -0.12/**-0.30** | 0.35 | **1.19/0.90** |
| PRECC (mm/day) | **-0.38** | **-0.81/-0.98** | 0.01 | **-0.31/-0.33** | **0.62** | **-1.57/-1.74** |
| PRECL (mm/day) | -0.06 | -0.25/**-0.88** | **-0.25** | 0.19/0.03 | **-0.27** | **2.76/2.64** |

"Table 2. Simulation differences due to physical parameterizations (CAM6α-1˚ minus CAM5-1˚) and high horizontal resolution (CAM6α-0.25˚ minus CAM6α-1˚), respectively. The values are averages over Sichuan and Northern China during 1980-2004."
==>
"Table 2. Simulation differences due to physical parameterizations (CAM6α-1˚ minus CAM5-1˚) and high horizontal resolution (CAM6α-0.25˚ minus CAM6α-1˚/CAM5-0.25˚ minus CAM5-1˚), respectively. The values are averages over Southwestern China, Northern China and Maritime continent during 1980-2004."

[Figure]

Added the discussion as below:

"Next we explore the differences in simulated variables due to higher horizontal resolution in CAM6 over the Maritime Continent (Figure 3). As seen in Table 2, the higher horizontal resolution in CAM6 not only increases the vertically-integrated total cloud cover over the Maritime continent, but also leads to more shortwave flux reaching the surface, which tends to release more latent heat. Both CAM5 and 6 versions with 0.25˚ resolution reduce the convective precipitation and increase the large-scale precipitation relative to 1˚ resolution, which leads to overestimation of total precipitation. The two CAM versions with higher resolution simulate a different vertically-integrated total cloud change (Table 2). With a fully coupled Community Climate System Model Version 4 (CCSM4), an earlier version of CAM (CAM4), Shields et al. [2016] have shown that higher horizontal resolution tends to decrease convective precipitation and increase large-scale precipitation. The moisture budget analysis shows that the meridional specific humidity eddy transport is the main factor leading to the bias over Maritime continent (Figure 14c)."

Minor Points

1) The authors need to be more explicit about what they mean when they refer to regions such as "Sichuan" or "southern China". I know figure 2c has boxes denoting regions, yet these boxes are not labeled anywhere in the figure or caption. Further, these regions are referred to in Figure 1 and it required efforts of my own to try to identify what the authors were referring to in reference to "Sichuan" etc. Explicitly defining these regions EARLY in the paper will go a long way towards improving the clarity of the paper.

Response:

Accepted. We mixed Sichuan province, "Sichuan box" and Southern China. We now adjusted as below. "Sichuan box" marked as "Southwestern China".

"Because of the large spatial heterogeneity, eight regions are selected to evaluate precipitation. Five domains shown as the purple boxes of Figure 2c are Tibet, Sichuan, Korea, Japan and the Maritime Continent. The other three are India, Northern China and Southern China."
==>
"Because of the large spatial heterogeneity, eight regions are selected to evaluate precipitation. Five domains shown as the purple boxes of Figure 2c are (1) Tibet: 27°N–37°N, 79°E–99°E; (2) Southwestern China: 28.5°N–35.5°N, 100°E–105°E; (3) Korea: 34°N–40°N, 124.5°E–129.5°E; (4) Japan: 31°N–43°N, 130°E–144°E; (5) the Maritime Continent: 9.75°S–19.75°N, 90°E–150°E. The other three are India, Northern China and Southern China. The "India" average is entirely within mainland India. "Northern China" and "Southern China", are also defined in Figure 2c, as the where "Northern China" north of the "Qin Mountain and Huai River" at 32.8°N, and "Southern China" south of this. The western boundary of "Northern China" and "Southern China" is a straight line named as "Hu-Huanyong Line" between Heihe (50.2°N, 127.5°E) and Tengchong (24.5°N, 98.0°E)."

2) Table one should include any difference in the time step between the 1 degree and 0.25 degree model. In addition, was the deep convective time scale adjusted in the 0.25 degree simulation versus the 1 degree simulation?

Response:
Accepted. We have added the corresponding information in Table one:

|  | CAM5-1º | CAM6α-1º | CAM6α-0.25º | CAM5-0.25º |
|---|---|---|---|---|
| timestep | 1800s | 900s | 900s | 1800s |

| The deep convective time scale | 3600s | 3600s | 3600s | 3600s |
|---|---|---|---|---|

Anonymous Referee #2

GENERAL COMMENTS

This is an interesting paper evaluating the performance of the CAM6 prototype in the Asian region at two horizontal resolutions and comparing against CAM5. The paper is reasonably well-written and the figures are clear. I have some concerns regarding the use of reanalyses data as a second benchmark, and it would helpful to clarify how some of the resolution comparisons have been made.
Finally, it would be helpful if the manuscript could be checked carefully by a native English speaker to remove numerous errors.

Response:
    Thank you for the time for reviewing and suggesting helpful revisions. We added the CPC data as your suggestion. And the English has been checked by a native English speaker.

SPECIFIC COMMENTS

page 2, line 25: Actually, GA6 is not the latest atmosphere model from the UK Met Office. Williams et al. (2017; https://agupubs.onlinelibrary.wiley.com/doi/full/10.1002/2017MS001115) state that the GC3 coupled configuration, which has GA7 as its basis, will form the basis for their CMIP6 submissions. There are substantial changes between GA6 and GA7. You can still quote the studies relating to GA6, but perhaps reword this paragraph.

Response:
    Thanks for the information, we adjusted the statement:
"For example, Global Atmosphere 6.0 (GA6), the latest atmosphere model from the UK Met Office, was used to study the interannual and intraseasonal precipitation variability over China [Stephan et al., 2018a and b]. GA6 includes a new dynamical core and updates various physical parameterizations [Walters et al., 2017]."
==>
"For example, the UK Met Office atmosphere model Global Atmosphere 6.0 (GA6) was used to study the interannual and intraseasonal precipitation variability over China [Stephan et al., 2018a and b; Walters et al., 2017a]."

page 3, line 1: Enhanced model resolution has not always been demonstrated as a means to reduced model biases, especially not in the tropics. For example, Johnson et al (2016; https://link.springer.com/article/10.1007%2Fs00382-015-2614-1) showed that increasing horizontal resolution was not a solution to the South Asian monsoon biases in the Met Office GA3 model and also stated, based on past studies, that "it is difficult to attribute the monsoon

improvement to any particular physics or resolution change in the atmosphere or ocean components."

Response:
"Enhanced model resolution has been demonstrated as a means to reduce model biases [Palmer, 2014; Yao et al., 2017; Chen et al., 2018]."
==>
"Generally, enhanced model resolution tends to reduce model biases [Palmer, 2014; Yao et al., 2017; Chen et al., 2018]. Nevertheless, Johnson et al. (2016) indicated that increasing horizontal resolution was not a solution to many South Asian monsoon biases in the Met Office Global Atmosphere 3.0 ( GA3) model."

Johnson, S. J., Levine, R. C., Turner, A. G., Martin, G. M., Woolnough, S. J., Schiemann, R., ... & Strachan, J. (2016). The resolution sensitivity of the South Asian monsoon and Indo-Pacific in a global 0.35 AGCM. *Climate Dynamics*, *46*(3-4), 807-831.

page 4, line 2: "time-varying observed sea surface temperatures and sea ice" - what is the time resolution of your forcing dataset? Is it monthly interpolated or actual daily mean values?

Response:
"time-varying observed sea surface temperatures and sea ice"
==>
"observed monthly sea surface temperature and sea ice from 1979 to 2005, which are linearly interpolated to obtain specified daily values"

Section 2.2: I am a little concerned that there is really only one observational rainfall dataset employed here. Rainfall from reanalyses is very dependent on the model physics, and is therefore not really a suitable benchmark. I realise that you may be constrained by the time period and horizontal resolution of your simulations, but it would be good to include more caveats on the APHRODITE data, particularly the potential lack of gauge observations in mountainous areas: how reliable are the values over Tibet, the Himalayas and the Maritime Continent islands? While I would not expect a detailed comparison between observational datasets in your study, some additional discussion on this aspect is warranted, instead of simply says in lines 18-20 of page 4 that you use the reanalyses as a benchmark, thereby implying that this is suitable. Similarly, there are other global datasets for surface air temperature from in situ measurements (such as CPC: https://www.esrl.noaa.gov/psd/data/gridded/data.cpc.globaltemp.html) that could be used as a second benchmark.

Response:
Agreed, APHRODITE might short of surface measurement over mountainous areas, we added CPC daily temperature as your suggestion. Therefore, we get two 'benchmark', while the reanalysis (temperature in JRA55 and daily precipitation from MERRA2) are used as additional 'data products' to evaluate model performance. A key point in our evaluation is that AMIP-style

model should not be expected to outperform the reanalysis as the latter is constrained by many other physical quantities.

"We adopted MERRA2 as an additional benchmark specifically for daily precipitation evaluation, because of the well-known large uncertainty among various observational datasets"
==>
"The APHRODITE data might be limited by the potential lack of gauge observations in mountainous areas [Zhao et al., 2015], and therefore we adopted MERRA2 as an additional data source specifically for daily precipitation evaluation. Note, however, that rainfall in the reanalysis products such as JRA55 and MERRA2 is dependent on the reanalysis model physics, and it is well-known that there is large uncertainty among various observational datasets"

We now also adopted the CPC daily temperature as a second benchmark.

The text in below was added in the manuscript:
"2.2.5 Climate Prediction Center (CPC) temperature
CPC global datasets for daily surface air temperature from in situ measurements with 0.5º × 0.5º resolution is used as a second benchmark [Chen et al., 2008]. CPC data is provided by the NOAA/OAR/ESRL (National Oceanic and Atmospheric Administration/Oceanic and Atmospheric Research/Earth System Research Laboratory) PSD (Physical Sciences Division), Boulder, Colorado, USA."

Data availability:
"The CPC datasets is from:
https://www.esrl.noaa.gov/psd/data/gridded/data.cpc.globaltemp.html. "

[Figure]

"First note that although sharing some common surface measurements, the two observational datasets (JRA55 and APHRODITE) have major differences over Tibet and southeast Asia regions (Figure 1a)."
==>
"First note that although sharing some common surface measurements, JRA55 and APHRODITE have major differences over the Tibetan Plateau and Southeast Asia regions (Figure 1a)"

Figures 1 and 2: Please confirm in the caption that you have interpolated the observations/reanalyses to the model's 1 degree grid, and the APHRODITE data to the JRA-55's 0.56 degree resolution, for panels (a) to (c) of these figures?

Response:
    Thanks for the concern, we confirmed that and added the text in below to the caption:

"All the data were interpolated to 1º resolution."

page 5, line 17: "...to fully capture the larger uncertainty of observational datasets" - see previous comment regarding reanalyses. I suggest "...as an estimate of the large uncertainty..." would be more appropriate.

Response:
        Thanks for the suggestion, we changed that:
"to fully capture the larger uncertainty of observational datasets"

==》

"as an estimate of the large uncertainty of observational datasets"

page 6, lines 13-14 and 23-24: An increase in horizontal resolution between 1 degree and 0.25 degrees is grossly insufficient to avoid the use of parameterization of clouds and convection by resolving those processes. Even in a prognostic scheme such as CLUBB, the microphysical processes are still parameterized: a prognostic increment to the condensate and rainfall is calculated by making assumptions about (i.e. parameterizing) the relationship between the thermodynamic variables and the microphysical process. Increasing resolution could change the local circulations, the thermodynamic variables and their distribution, and may make them more realistic if the processes driving those are resolved better, but this is not actually resolving the precipitation process itself better. Ultimately, as you note in line 17, the partition between the large-scale and convective rainfall in any model (including reanalyses), and how this changes with resolution, will depend on the parameterization schemes employed.
There are also several statements in this section that imply that convection parameterizations are only there to mop up instability. I do not think that this is true. They should be representing the effects of sub-gridscale convective processes (as opposed to subgridscale stratiform cloud processes, which are handled by the large-scale cloud and precipitation parameterization) on the environment in the gridbox.
Please consider rewording these misleading sentences.

Response:
        Thanks for your insightful comments. We had rewritten these sentences:

Page 6, line 13-14:
"One of the reason that the simulated rainfall intensity is expect to improve when the model run at higher resolution is small-scale processes actually simulated as opposed to to parameterizations would produce more rainfall [Kopparla et al., 2013]."
==>
"One of the reasons why the simulated rainfall intensity is expected to improve when the model is run at higher resolution is that the variance of sub-grid scale humidity and thermodynamics drops, and the parameterized sub-grid scale processes (such as sub-grid scale turbulence with CLUBB) are better separated into regimes [Kopparla et al., 2013]."

Page 6, line 22-24:

"Higher horizontal resolution models tend to simulate higher vertical velocities [Gettelman et al., 2018] and a lower ratio of convective to total rainfall, because a larger fraction of precipitation can be resolved as the consequences of large-scale flow, limiting the need of invoking convective schemes."
==>
"Higher horizontal resolution models tend to simulate higher vertical velocities [Gettelman et al., 2018] and a lower ratio of convective to total rainfall. A larger fraction of precipitation can be resolved as the consequence of large-scale flow, limiting the need to invoke sub-grid convective schemes. Besides, increasing resolution better resolves topographic and surface effects, and separates regimes as sub-grid scale variance is reduced, particularly in the thermodynamic variables."

The statements on the instability:
"The ratio of convective and large-scale precipitation is a useful diagnostic, because both convective activity and large-scale instability can lead to precipitation in this model. Most atmospheric models use the convective parameterizations to balance large-scale thermal instabilities, and not to derive grid-scale microphysical and precipitation processes."
==>
"The ratio of convective to large-scale precipitation is a useful diagnostic, because both convective activity and large-scale instability can lead to precipitation in this model. Most atmospheric models use convective parameterizations to represent the effects of sub-grid scale convective processes, with reduced complexity microphysics"

"CAM6 estimate the shallow convective precipitation from explicit prognostic calculation rather than diagnostically estimate from large-scale instabilities."
==>
"CAM6 estimates shallow convective precipitation from the prognostic calculations in CLUBB (which have memory between timesteps of turbulent motion) rather than diagnostically representing the effects of sub-grid scale convective processes at each location and timestep."

"In contrast, the convective parameterization with a timescale produces mass flux and precipitation at a defined rate and consumes instability."
==>
"In contrast, the deep convective parameterization has a timescale that produces mass flux and precipitation at a defined rate. Note that as the timestep gets shorter, the mass flux and precipitation over a timestep will decrease, which is the major reason for the decrease in deep convective precipitation in CAM6."

page 6, lines 16-21 and Figure 4: Despite your caveat that JRA-55 is providing a partition that is model-dependent, its inclusion in Figure 4 implies that you are considering it as a benchmark for the model comparison. I recommend that you remove this and only consider the comparison between CAM versions.

Instead, you could include in Figure 4 some evidence that the change in timestep affects the partitioning in the way that you assert in lines 26-31 (by showing results from CAM6-1 with a 10 minute timestep, perhaps?).

Response:
Thanks for the good idea. We adjusted the statement and refer a paper to explain the effects of  timestep. We just keep it without comparing it with model in the text.

"The reanalysis product JRA55 also provides a decomposition of convective and large scale precipitation. (Figure 4a). Note that the convective and large-scale rainfall in JRA55 is not really observations and the partitioning is highly dependent on the JRA55 model assumptions. The ratio of convective (PRECC) to large-scale (PRECL) precipitation is greater over the ocean than over the land, as expected. CAM6α-1˚ has a larger ratio over the tropics, compared to CAM5-1º and JRA55 (Figure 4c). The CAM6α-0.25˚ simulated ratio is closer to JRA55, due to higher spatial resolution in both datasets (Figure 4d).
...
In addition, the compensation above is a feature of the physical parameterization suite in CAM due to timescale. CLUBB and the prognostic cloud microphysics are run whenever large scale condensation that process removes all liquid supersaturation instantaneously. In contrast, the convective parameterization with a timescale produces mass flux and precipitation at a defined rate and consumes instability. The large scale condensation (including shallow convection and cloud microphysics) does more as the time step changes, while the deep convective parameterization does less. The high-resolution model has a shorter timestep (10 minutes vs. 30 minutes)."
==>
"The reanalysis product JRA55 also provides a decomposition of convective and large-scale precipitation. (Figure 4a). Note that the convective and large-scale rainfall in JRA55 is not from observations and the partitioning is highly dependent on the JRA55 model assumptions. Figure 4 illustrates the ratio of convective (PRECC) to large-scale (PRECL) precipitation. This ratio (PRECC/PRECL) is greater over the ocean than over the land, as expected. CAM6α-1˚ has a larger ratio over the tropics, compared to CAM5-1º. JRA55 (Figure 4c) has a similar ratio, though the thresholds and model differences make a direct quantitative comparison inappropriate. CAM6α-0.25˚ simulated a lower ratio than CAM6α-1˚ (Figure 4d).
...
The compensation above is a feature of the physical parameterization suite in CAM due to timescale. Large-scale liquid condensation by the resolved scale cloud schemes (CLUBB and microphysics), instantaneously condenses all vapor in excess of liquid saturation to cloud liquid. In contrast, the deep convective parameterization has a timescale that produces mass flux and precipitation at a defined rate. Note that as the timestep gets shorter, the mass flux and precipitation over a timestep will decrease, which is the major reason for the decrease in deep convective precipitation in CAM6. The large-scale condensation (including shallow convection and cloud microphysics) does more as the time step changes, while the deep convective parameterization does less [Gettelman et al., 2018]."

page 7, lines 1-13, and Figure 5: Please confirm that you have compared all of these on the same 1 degree grid resolution? This is particularly important for the RX1day and R10 statistics that are measured using threshold values against the intensity distribution, which itself will depend on the resolution of the data (even for the observations).

Response:
      Yes, we double checked. We realised it is unclear to the reader on the resolution. We added the text in below to the caption: "All the data were interpolated to 1º resolution."

page 8, line 4: "higher frequency fluctuations" - are these really higher frequency fluctuations? It is still an interannual variation, albeit of seasonal mean values.
Also, given the known influences of ENSO on boreal summer monsoon rainfall over Asia, it would be interesting to compare the regression of JJA mean rainfall against ENSO in Figure 7, as well as that of the annual mean rainfall.

Response:
"Next, we examine precipitation variability associated with higher-frequency fluctuations due to"
==>
"Next, we examine seasonal precipitation variability associated with the East Asia Summer Monsoon (EASM)."

It is good idea to compare the regression of JJA mean rainfall against ENSO. We did so.

[Figure]

[Figure]

Regression JJA Precipitation with ENSO (1980-2004)

-0.016  -0.012  -0.008  -0.004  -0.002  0.002  0.004  0.008  0.012  0.016

"**Figure 7. Annual mean precipitation regressed onto the observed ENSO index (mm/day) for 1980-2004**."

==>

"**Figure 7. Regression of annual mean (a-f) / summer (JJA, g-l) precipitation onto the observed ENSO index (mm/day) for 1980-2004.**"

"Figure 7 shows the regression coefficients between annual mean precipitation and ENSO (the cold tongue index used as the ENSO index in this study) [Deser and Wallace, 1987]. Both 1º CAM model (Figure 7d and e) capture the observed wetting anomaly over Pakistan and Afghanistan and drying anomaly over Indonesia. However, we find that the drying tendency over Southern China during El Nino years (the upper row of Figure 7) is completely missing in CAM5 but starts to emerge in CAM6α-1º version and gets better in CAM6α-0.25º version. Our results here thus call into questions of the fidelity of previous ENSO studies on hydroclimate over Southern China using CAM5."

==>

"Figure 7a-f shows the regression coefficients between annual mean precipitation and ENSO. Where the ENSO index is the cold tongue index following Deser and Wallace [1987]. Both 1º CAM5 and CAM6a (Figure 7d and e) capture the observed moist anomaly over Pakistan and Afghanistan and dry anomaly over Indonesia. However, we find that the drying tendency over Southern China during El Nino years (the upper row of Figure 7) is completely missing in CAM5 but starts to emerge in CAM6α-1º version and gets better in CAM6α-0.25º version. Figure 7g-l show similar patterns but with stronger correlation. Our results here thus call into questions of the fidelity of previous ENSO studies on hydroclimate over Southern China using CAM5."

Section 4.3 and Figures 11 and 12: Similar to my previous comment regarding Figure 5, please confirm that everything has been reduced to the same horizontal resolution before doing this analysis.

Response:
Yes, we harmonized the resolution to 1 deg before do the regional average and we added the text in the caption:
"All the data were interpolated to 1º resolution before regional average."

Further, you have suddenly taken the mean of the observations and reanalyses here, in what is perhaps the hardest test for the models. It would be advisable to state or show how these distributions vary between APHRODITE and the two reanalyses - is this really captured with the one standard deviation? There are only three datasets, so why not show the envelope as shading instead (and APHRODITE as the solid line)?

Response:
Yes, we agree with you. The standard deviation can be misleading with only three datasets. We now show the envelope instead shading.

[Figure]

"Figure 11. … Black lines (shading) for the mean (one standard deviation) of APHRODITE, JRA55 and MERRA2. ..."
==>
"Figure 11. … Black lines (shading) for the APHRODITE (the maximum and minimum of APHRODITE, JRA55 and MERRA2). ..."

page 9 line 16: The CAM6-0.25 also looks worse for heavy rainfall (than CAM6-1) for the Sichuan and Tibet regions.

Response:

Yes, that is why we made the following statement:
"Higher horizontal resolution in CAM6 (green) simulates better intensities over the Maritime Continent (Figure 12e), but the results degrade for the heaviest precipitation events over India, Northern and Southern China (Figure 12f-h)."

page 10, lines 3-5: "High-resolution simulations..." - where is your evidence for this statement? Even if it is true (of which I am not sure), there is also horizontal moisture advection which will be different at higher resolution.

Response:
        Sorry for misunderstanding, we adjusted the statement and refer the evidence to the Table 2:

"High-resolution simulations tend to decrease the energy used for the convective process (decreased latent heat flux and sensible heat flux), and thus decrease the convective precipitation."
==>
"Higher resolutions in CAM6 and CAM5 both decrease the surface latent heat flux and convective precipitation over Southwestern China (Table 2)."

page 10, line 6: Presumably the increased downward solar radiation in CAM6 is related to the improved diurnal cycle? Please state this, if it is the case.

Response:
        It is possible, but without sub-daily data, we are not able to confirm this. That text is just based on the Table 2:
"New physics parameterizations in CAM6 simulate stronger solar flux reach the surface."
==>
"Newer physics parameterizations in CAM6α simulate a stronger solar flux reaching the surface in northern China (Table 2). This may be due to improvements in the diurnal cycle of precipitation."

Section 5: Overall, I do not find this section convincing, nor does it add much to the findings of the study. You make several statements about the impact of the new physics in CAM6 on the balance of processes and the large-scale/convective rainfall partition that are speculative and not supported by evidence. I would suggest that you remove this section and Figure 14 (and the associated bullet point (4) on page 11).

Response:
        We agree with you, that part is unclear to the readers. Actually, that section is used to attribute improvements to either new physics module or higher resolution. We contrast climate variables quantitatively in Table 2. Those statements are all based on Table 2 actually. So we added the text in the beginning of Section 5:

"We attempt to attribute changes to either physical parameterizations or resolution. Additionally, we investigate whether the improvement due to resolution is dependent on CAM version. Table 2 illustrates simulation differences due to physical parameterizations (CAM6α-1˚ minus CAM5-1˚) and higher horizontal resolution (CAM6α-0.25˚ minus CAM6α-1˚ and CAM5-0.25˚ minus CAM5-1˚, respectively)."

page 11 line 17: "...better performance over Sichuan basin..." - is this true? It looks worse than CAM6 at 1 deg in Figure 12.

Response:
It might not be clear. We are talking about the PDF of daily P (refer to Figure 11), not daily P as a function of percentile (refer to Figure 12).

"higher horizontal resolution leads to better performance over Sichuan basin (close to the edge of Tibet plateau)"
==>
"higher horizontal resolution in CAM6 leads to better performance on the frequency distribution of daily precipitation over Southwestern China (close to the edge of Tibet Plateau)"

TECHNICAL CORRECTIONS
[Note that, in addition to the points raised below, there are numerous wording and grammar issues that require careful editing by a native English speaker]
page 2 line 11: "regarding to" -> relating to

Response:
Thanks, corrected.

page 3, line 27: Please expand CLUBB, or at least mention which part of the model physics this relates to.

Response:
Thanks, we expanded: CLUBB (Cloud Layers Unified By Binormals).

page 6, line 27: "...whenever large scale condensation that process removes all liquid..." - I do not understand this sentence.

Response:
"CLUBB and the prognostic cloud microphysics are run whenever large scale condensation that process removes all liquid supersaturation instantaneously."
==>
"Large-scale liquid condensation by the resolved scale cloud schemes (CLUBB and microphysics), instantaneously condenses all vapor in excess of liquid saturation to cloud liquid."

page 8, line 16: "...edge of the positive correlation is less more northward..." - more, I think.

Response:
"...edge of the positive correlation is less more northward..."
==>
"...edge of the positive correlation more northward..."

Figure 10 caption: This is not just within Southern China.

Response:
"Southern China"
==>
"Part of Asia"

page 9, line 1: "which is missing the persistent in CAM5-1 (the jumping cliff over July to August..." ??

Response:
"which is missing the persistent in CAM5-1˚ (the jumping cliff over July to August, Figure 10d)."
==>
"while CAM5-1º illustrates persistence from June to July only (the area of continuous yellow shading in Figure 10d less than those of CAM6). "

Table 2 - you have not referred to this table anywhere in the manuscript. It may not be needed if you remove section 5.

Response:
    We realized that we did not refer it in the text, we referred it now. That table is actually very important to explain the simulation differences due to resolution or new physical module. Please refer to the responses to the last few Minor comments.

**CAM6 simulation of mean and extreme precipitation over Asia: Sensitivity to upgraded physical parameterizations and higher horizontal resolution**

Lei Lin[1], Andrew Gettelman[2], Yangyang Xu[3], Chenglai Wu[4], Zhili Wang[5], Nan Rosenbloom[2], Susan C. Bates[2], Wenjie Dong[1]

[1]School of Atmospheric Sciences and Guangdong Province Key Laboratory for Climate Change and Natural Disaster Studies, Sun Yat-sen University, Zhuhai, Guangdong, China
[2]National Center for Atmospheric Research, Boulder, Colorado, USA
[3]Department of Atmospheric Sciences, College of Geosciences, Texas A&M University, College Station, Texas, USA
[4]International Center for Climate and Environment Sciences, Institute of Atmospheric Physics, Chinese Academy of Sciences, Beijing, China
[5]State Key Laboratory of Severe Weather and Key Laboratory of Atmospheric Chemistry of CMA, Chinese Academy of Meteorological Sciences, Beijing, China

*Correspondence to*: Wenjie Dong (dongwj3@mail.sysu.edu.cn)

**Abstract.** The Community Atmosphere Model version 6 (CAM6) released in 2018, as part of the Community Earth System Model version 2 (CESM2), is a major upgrade over the previous CAM5 that has been used in numerous global and regional climate studies. Since CESM2/CAM6 will participate in the upcoming Coupled Model Intercomparison Project phase 6 (CMIP6) and is likely to be adopted in many future studies, its simulation fidelity needs to be thoroughly examined. Here we evaluate the performance of a developmental version of the Community Atmosphere Model with parameterizations that will be used in the version 6 (CAM6α) with a default 1º horizontal resolution (0.9º × 1.25º, CAM6α-1º) and a high resolution configuration (approximately 0.25º, CAM6α-0.25º), against various observational and reanalysis datasets of precipitation over Asia. CAM6α performance is compared with CAM5 at default 1º horizontal resolution (CAM5-1º) and a high-resolution configuration at 0.25º (CAM5-0.25º). With the prognostic treatment of precipitation processes and the new microphysics module, CAM6α is able to better simulate climatological mean and extreme precipitation over Asia, to better capture the heaviest precipitation events, to better reproduce the diurnal cycle of precipitation rates over most of Asia, and to better simulate the probability density distributions of daily precipitation over Tibet, Korea, Japan and Northern China. Higher horizontal resolution in CAM6α improves the simulation of mean and extreme precipitation over Northern China, but the performance degrades over the Maritime continent. Moisture budget diagnosis suggests that the physical processes leading to model improvement are different over different regions. Both upgraded physical parameterizations and higher horizontal resolution affect the simulated precipitation response to internal variability of the climate system (e.g. Asia monsoon variability, ENSO, PDO), but the effects vary across different regions. For example, higher horizontal resolution degrades the model performance in simulating precipitation variability over Southern China associated with the East Asia summer monsoon. In contrast, precipitation variability associated with ENSO improves with upgraded physical

parameterizations and higher horizontal resolution. CAM6α-0.25º and CAM6α-1º shows an opposite response to the PDO over Southern China. Basically, the response to increases in horizontal resolution is dependent on CAM version.

**1 Introduction**

[revised manuscript text omitted]

$$\overline{P} + \overline{<\partial_x(uq)>} + \overline{<\partial_y(vq)>} + \text{Re}\,s = \overline{E} \quad ,$$

(1)

where P is precipitation, q is specific humidity, u (v) is zonal (meridional) wind and E is evaporation into the atmosphere. <X> is a mass-weighted vertical integral and $\overline{X}$ denotes a temporal average. The horizontal advection can be further decomposed into the stationary and transient terms based on:

$$X = \overline{X} + X' = [\overline{X}] + \overline{X}^* + X' \quad ,$$

(2)

删除[l]: **Improvement**

删除[l]: **vs. Sichuan**

删除[l]: We

删除[l]: the

删除[l]: Sichuan and

删除[l]: here

删除[l]: module

删除[l]: both simulated precipitation

删除[l]: better

删除[l]: those two regions

删除[l]: .

[revised manuscript text omitted]

删除[l]: :
删除[l]: ,
删除[l]: :
删除[l]: ,
删除[l]: :
删除[l]: ,
删除[l]: :
删除[l]: ,
删除[l]: :
删除[l]: ,
删除[l]: :
删除[l]: ,
删除[l]: :
删除[l]: ,
删除[l]: ,
删除[l]: :
删除[l]: ,
删除[l]: :
删除[l]: ,
删除[l]: :
删除[l]: ,
删除[l]: :
删除[l]: ,
删除[l]: :
删除[l]: ,

Seager, R., Naik, N., & Vecchi, G. A.: Thermodynamic and dynamic mechanisms for large-scale changes in the hydrological cycle in response to global warming, Journal of Climate, 23(17), 4651-4668, 2010.

[revised manuscript text omitted]

**Figure 6Regression of annual l mean precipitation (mm/day) onto observed PDO indices for 1980-2004.**

删除[l]: **. Annua**

删除[l]: **) regresse**

删除[l]:

[Figure]

[Figure]

Regression JJA Precipitation with ENSO (1980-2004)

[Figure]

Regression Annual Precipitation with ENSO (1980-2004)

**Figure Regression of annual al me(a-f) / summer (JJA, g-l) an precipitatied onto the observed ENSO index (mm/day) for 1980-24.**

[Figure]

Regression JJA Precipitation with EASMI (1980-2004)

删除[l]: **7. Annu**

删除[l]: **on regress**

删除[l]: **00**

删除[l]:

[Figure]

Regression JJA Precipitation with EASMI (1980-2004)

**F**igure mer (JJA) precipitated onto EASMI (mm/day) for 1980-2004.

删除[l]:  **8. Sum**

删除[l]:  **ion regres**

[Figure]

**Figure 9. The difference of summer minus winter (JJA - DJF) precipitation (mm/day) between APHRODITE and a) JRA55, b) MERRA2, c) CAM5-1°, d) CAM6α-1° and e) CAM6α-0.25°, respectively. The values at the lower left of every panel indicate the root mean squared difference (RMSD) relative to APHRO. Grid points in panel (c)-(e) are stippled when the absolute value of difference between model and APHRODITE is larger than that between JRA55 and APHRODITEDITE.**

[Figure]

**Figure 10. Annual cycle of regional mean precipitation rate (mm/day) for 1980-2004 with** **Part of Asia** **ina (between 100˚E and 125˚E): a) APHRODITE, b) JRA55, c) MERRA2, d) CAM5-1˚, e) CAM6α-1˚ and f) CAM6α-0.25˚**

删除[l]:  **hin Southern Ch**

[Figure]

删除[l]:

[Figure]

**Figure 11.** Frequency distribution of daily precipitation (mm/day) over a) Tibet,Southwestern Chinahuan, c) Korea, d) Japan, e) maritime continent (see boxes in Fig2c 2.c), f) India, g) Northern China and h) Southern Chare ina for 1980-2004. Black lines (shading) for APHRODITE (the maximum and minimum on) of APHRODITE, JRA55 and ME)RRA2. Red solid liare nes for CAM5-1º; blue solid lines for CAM6α-1º; green solid lines for CAM6α-0; purple solid lines for CAM5-0.25º.2All the data were

删除[l]: **b) Sic**

删除[l]: **ure**

删除[l]: **the mean (one standard deviati**

interpolated to 1° resolution before regional

删除[1]:
[Figure]

[Figure]

**average.5°**

[Figure]

**Figure 12. Similar to Figure 11, but for daily precipitation rates (mm/day) as a function of percentile.**

[Figure]

Figure 13. Diurnal cycle of precipitation in June (6-year average from 1999-2004) of (a) TRMM, (b) CAM5-1°, (c) CAM6α-1° and (d) CAM6α-0.25°. The local time peak of the diurnal cycle is shown in color on the color wheel. The intensity of the color is the

[Figure]

[Figure]

删除[l]: n

amplitude of precipitatio

[Figure]

**Figure 14. Moisture budget oveSouthwestern China, and b) Northern Cand c) Maritime continent hina (see the boxes of Figure 2c) for 1980-2004, including precipitation, evaporation, zonal and meridional moisture flux convergence, residual and (right to dashed line) major processes contributing to moisture flux converg**

5    **ence.**

删除[l]:  **r a) Sichuan**

---

## Author Response (AR2)

I thank the authors for their revisions to the manuscript. While these have largely improved the submission, unfortunately there are now some further important revisions that should be made as a result of their changes. Mainly, while the authors have followed the excellent suggestion made by Reviewer #1 to include CAM5-0.25, results from this are not shown in many of the key figures, for no apparent reason. These really should be added, and the findings discussed, particularly to support the conclusion added at the end of the Summary and the Abstract, namely that "the response to increases in horizontal resolution is dependent on CAM version."

Response:

We appreciate the meticulous comments from the reviewer, and we agree that more results and discussion related to CAM5-0.25º simulation should be added. But we also emphasize that the main focus of this paper is on the CAM6 sensitivity to higher resolution and physics (as apparently stated in the title), because the research community is likely to use CAM6 a lot in the next 5-10 years, while the application of CAM5 and earlier models will gradually sunset.

Therefore, we have taken the reviewers's suggestion to include CAM5-0.25º analysis in all of the figures. The results support one of the conclusions as highlighted by the reviewer - "the response to increases in horizontal resolution is dependent on CAM version."

Specific comments:

Page 4 lines 19-20: "The dataset was created by collating rain gauge measurements across Asia with gridded daily data that contains a dense network of daily rain-gauge data for Asia." ? Unclear.

Response:

Now rephrased as:

"The dataset was created by collating a dense network of rain gauge daily measurements across Asia."

Page 5 lines 18-19: As mentioned by Reviewer #1, the reader may be unclear where Sichuan province is, so please forward-reference the appropriate box in Figure 2. Further, although the purple boxes are clearly described on page 6, they should also be mentioned in the caption for Figure 2. In order to make this easier for the reader, perhaps you could consider drawing each box in a different colour on Figure 2 and then labelling them with names (rather than latitude/longitude boundaries) in the caption?

Response:

"Sichuan province" means the administrative division, not that box in Figure 2. We have renamed that box as "Southwestern China".

And yes, we used different color for different boxes and updated the Figure 2 caption based on the suggestion.

"Five domains are shown as colored boxes in Figure 2c: (1) Tibet: 27°N–37°N, 79°E–99°E; (2) Southwestern China: 28.5°N–35.5°N, 100°E–105°E; (3) Korea: 34°N–40°N, 124.5°E–129.5°E; (4) Japan: 31°N–43°N, 130°E–144°E; (5) the Maritime Continent: 9.75°S–19.75°N, 90°E–150°E. Only the land within these boxes is considered in this study. The other three are India, Northern China and Southern China."

Page 6 lines 15-16: Please clarify that you are (presumably) only considering the land within these boxes. This is particularly important for the Maritime Continent region, and especially for the analysis in section 5, Table 2 and Figure 14.

Response:

Added "Only the land within these boxes are considered in this study."

Sections 3 and 4.1, 4.2, and Figures 1, 2, 4, 5, 6, 7, 8, 9 and 10: Why is CAM5-0.25 not included in these figures and this analysis? The discussion of this configuration in other sections is out of context if it is omitted here. Further, it would be very helpful to show how the influence of resolution as shown for CAM6 differs for CAM5. Please add the equivalent panels/values to these figures. For Figures 6, 7, 8 and 10 you could remove MERRA2 if you prefer a 6-panel plot. Also, to highlight the influence of the CAM5/CAM6 change and those relating to resolution, plotting differences between these (rather than differences against the benchmark) would perhaps be helpful.

Response:

We add the CAM-0.25º results following the reviewer's suggestion. Concerned that there are so many plots already and due to space limitation, we have not put the differences CAM5 and CAM6 between in this version. But we can include those in the supplement file if the reviewer wants us to.

Page 7 lines 13-15 and Figure 4: I still maintain that JRA55 should not be included in this Figure or this discussion of convective/large-scale partitioning - especially now that it could be replaced by results from CAM5-0.25.

Response:

We removed JRA55, and replaced it with CAM5-0.25º as suggested.

Figure 5: Please add values from CPC to panel (a).

Response:

Added.

Page 9 line 3: Please add a sentence explaining to the reader what the other panels on Figure 7 show and why you added them.

Response:

"Figure 7g-l show similar patterns but with stronger correlation."

==>

"The influence of ENSO on boreal summer monsoon rainfall over Asia is well known. Figure 7h-n shows the regression coefficients between JJA precipitation and ENSO. Those patterns are similar when the analysis is done with annual mean precipitation but with a weaker correlation of annual precipitation and ENSO."

Page 9 lines 9-12: Given the plethora of monsoon indices in the literature, please add an explanation as to what the DNS monsoon index is

and how it is calculated.

Response:

"The East Asia Summer Monsoon Index (EASMI) is a unified Dynamic Normalized Seasonality (DNS) monsoon index defined by Li and Zeng [2002, 2003]. This index is based on the intensity of the monthly mean wind field and can be used to depict both the seasonal cycle and inter-annual variability of the EASM. The EASMI summer mean is considered here."

==>

"The East Asia Summer Monsoon Index (EASMI) is a unified Dynamic Normalized Seasonality (DNS) monsoon index defined by Li and Zeng [2002, 2003].

$$EASMI = \frac{\left\| \overline{V_1} - V_i \right\|}{\left\| \overline{V} \right\|} - 2$$

Where $\overline{V_1}, V_i$ are the January climatological and monthly wind vectors for a grid, respectively, and $\overline{V}$ is the mean of January and July climatological wind vectors for the same gird. The constant 2 on the right-hand side of the formula is the determinant criterion. The double vertical line indicates the normalized value. EASMI can be used to depict both the seasonal cycle and inter-annual variability of the EASM. The EASMI summer mean is considered here."

Page 10 lines 11-12: "CAM6 improves the light rainfall over Korea, Japan and Northern China (not shown)." - it is shown; the original first part of this sentence ("The large-scale precipitation in") has been removed, so maybe you are trying to say something different with this sentence?

Response:

Your are right, we make a mistake:

"CAM6 improves the light rainfall over Korea, Japan and Northern China (not shown)."

==>

"CAM6 improves the light large-scale precipitation over Korea, Japan and Northern China (not shown)."

Page 10 lines 16-17: Isn't the first part of this (better over the Maritime Continent) also true for CAM5-0.25, but for the latter configuration the higher resolution is slightly better than the lower resolution for India, Northern and Southern China?

Response:

Yes, we added.

"Higher horizontal resolution in CAM6 (green) simulates better intensities over the Maritime Continent (Figure 12e), but the results degrade for the heaviest precipitation events over India, Northern and Southern China (Figure 12f-h)."

==>

"Higher horizontal resolution in CAM6 and CAM5 (green and purple) simulates better intensities over the Maritime Continent (Figure 12e), but the results of CAM6 (CAM5) degrade (upgrade slightly) for the heaviest precipitation events over India, Northern and Southern China (Figure 12f-h)."

5 Figure 13: Please show and discuss results from CAM5-0.25 as well - it may reinforce your point about the influence of model physics (i.e. CLUBB), rather than resolution, in improving the diurnal cycle.

Response:

CAM5-0.25 is now shown and discussed: "CAM5 with higher resolution improves the diurnal cycle only slightly".

Table 2: Shouldn't the labelling of the fifth column be the same as that for the third and seventh columns?

Response:

Yes, sorry for that mistake, corrected.

Page 11 line 12: "less moisture" - do you mean less vertically-integrated humidity?

Response:

Yes, we changed "less moisture" to "less vertically-integrated humidity".

Page 11 line 15: Also higher resolution in CAM6/CAM5 leads to an opposite change in total cloud amount over Southwestern China.

Response:

"However, higher resolution leads to an opposite change in surface sensible heat flux between CAM5 and CAM6$\alpha$ (by 12.7 and -7.4 W/m$^2$,
25 respectively)"
==>
"However, higher resolution leads to an opposite change in surface sensible heat flux (total cloud amount) between CAM5 and CAM6$\alpha$ by 12.7 and -7.4 W/m$^2$ (-3.9 and 0.5 %), respectively."

30 Page 11 line 19: The lack of increase in PRECC despite increased FSDS and LHFLX may be related to the decrease in INT_Q?

Response:

INT_Q could be a reason but two CAM version shown the different change. Higher resolution in CAM6 shown the increased INT_Q.

35 Page 12 line 5: Isn't the largest difference (relative to JRA55) in the budget for southwestern China in the zonal moisture flux convergence term?

Response:

Thanks - you are right, we checked the values and corrected the statement.

Page 12 line 11: Remove "in CAM6" from this sentence.

Response:
Removed.

Page 12 lines 11-17: Please clarify the similarities and differences between the resolution impacts in CAM5 and CAM6, e.g. they have opposite changes in total cloud (presumably related to the larger increase in FSDS and LHFLX in CAM5?) and in downwelling clear-sky solar flux.

Response:
"As seen in Table 2, the higher horizontal resolution in CAM6 not only increases the vertically-integrated total cloud cover over the Maritime continent, but also leads to more shortwave flux reaching the surface, which tends to release more latent heat. Both CAM5 and 6 versions with 0.25˚ resolution reduce the convective precipitation and increase the large-scale precipitation relative to 1˚ resolution, which leads to overestimation of total precipitation. The two CAM versions with higher resolution simulate a different vertically-integrated total cloud change (Table 2)."
==>
"As seen in Table 2, the higher horizontal resolution in CAM6 not only increases the vertically-integrated total cloud cover over the Maritime continent, but also leads to more shortwave flux reaching the surface, which tends to release more latent heat. Both CAM5 and 6 versions with 0.25˚ resolution reduce the convective precipitation and increase the large-scale precipitation relative to 1˚ resolution, which leads to overestimation of total precipitation. The two 0.25˚ resolution CAM versions with higher resolution simulate the same surface air temperature, surface energy terms (surface latent and sensible heat flux, downwelling solar flux and longwave flux at surface) and vertically-integrated humidity change but simulate a different vertically-integrated total cloud change and downwelling clear-sky solar flux at surface (Table 2)."

Page 12 lines 19-20: Isn't it the zonal term that dominates?

Response:
Both $-\left[\overline{q}\right]\overline{\partial_x u^*}$ and $-\left[\overline{q}\right]\overline{\partial_y v^*}$ play an role, so we said $\left[\overline{q}\right]$ is the main factor.

Page 12 and Figure 14: I still feel dissatisfied with this section. The moisture budget analysis is not comprehensive for any of the regions, and particularly now that the Maritime Continent region has been included. There is ample literature suggesting that many GCMs show sensitivity to resolution over this region, relating to the representation of topography and coastlines of the islands. Johnson et al. (2016) carried out moisture budget analysis for the Maritime Continent region as a whole (including the ocean) that suggested the increased resolution altered the budget and the rainfall between the southern (windward) and northern (leeward) parts of the region. While you may not wish to go into great detail for this particular region in this study, a little more discussion of the moisture budget regarding the changes

with model physics and resolution, and reference to existing literature, would be helpful.

Response:

5    We can close the moisture budget by checking the residual term (now shown in Figure 14). It would be fine.

And yes, we agree it would be helpful with more discussion:

"Note that the analysis in this study is focused on the Maritime continent land area only. *Johnsan et al.* [2016] carried out a moisture budget analysis for the whole Maritime continent (including the ocean) and found that increased resolution causes increased moisture convergence and precipitation on the windward (southern) side of the orography, which leads to decreased moisture availability on the

10   leeward (northern) side, reducing precipitation."

Page 13 Summary: Other than a final sentence stating that the influence of resolution depends on the CAM model version, this section largely fails to draw out the details of this dependence (largely because CAM5-0.25 is omitted from many of the Figures). Clarifying this dependence would set a better context for the results from CAM6.

15   Response:

Thanks. We have now put more figures and discussions related to CAM5-0.25 into the manuscript. Please see the marked-up file for the details.

[revised manuscript text omitted]